nanotechnology

resveratrol, core–shell nanoparticles, drug delivery, bioavailability, antioxidant activity

**Author for correspondence:**
Mausumi Ganguly
e-mail: ganguly_mausumi@rediffmail.com

This article has been edited by the Royal Society of Chemistry, including the commissioning, peer review process and editorial aspects up to the point of acceptance.

# Resveratrol-loaded chitosan–pectin core–shell nanoparticles as novel drug delivery vehicle for sustained release and improved antioxidant activities

Shruti Sarma, Sangeeta Agarwal, Pranjal Bhuyan, Jnyandeep Hazarika and Mausumi Ganguly

Cotton University, Guwahati 781001, Assam, India

MG, 0000-0003-2267-8835

Resveratrol, chemically known as 3,5,4'-trihydroxy-trans-stilbene, is a natural polyphenol with promising multi-targeted health benefits. The optimal therapeutic uses of resveratrol are limited due to its poor solubility, rapid metabolism and low bioavailability. To address the issues, we have encapsulated resveratrol inside the nanosized core made of chitosan and coated this core with pectin-shell in order to fabricate a drug delivery vehicle which can entrap resveratrol for a longer period of time. The core–shell nanoparticles fabricated in this way were characterized with the help of Fourier transform infrared spectrometer, field-emission scanning electron microscope, field-emission transmission electron microscopy/ selected area electron diffraction, high-resolution transmission electron microscope, dynamic light scattering and zeta potential measurements. *In vitro* drug release study showed the ability of the core–shell nanoparticles to provide sustained release of resveratrol for almost 30 h. The release efficiency of the drug was found to be pH dependent, and a sequential control over drug release can be obtained by varying the shell thickness. The resveratrol encapsulated in a nanocarrier was found to have a better *in vitro* antioxidant activity than free resveratrol as determined by 2,2-diphenyl-1-picrylhydrazyl (DPPH) radical scavenging method. This work finally offers a novel nano-based drug delivery system.

# 1. Introduction

One of the prime attributes of an efficient drug-delivery vehicle is the controlled drug release rate. To increase the efficiency and therapeutic activity, the drug-delivery system should be made trigger dependent with sustained release behaviour. For this, various multifunctional nanoparticles with multiple triggers and complex structures have been developed [1,2]. Nowadays, nano-carriers are being frequently used for drug delivery. These nano-carriers can decrease the frequency of administration while maintaining the steady and effective concentration of the drug at the target site [3,4]. Several nanoparticle-based products for diagnostics and therapeutics have been clinically approved and many of them are currently under clinical trials [5]. During the last few decades, the main focus was on the development of biodegradable nanoparticles for effective drug delivery [6]. Among various classes of nanoparticles, the core–shell is the most promising class with various biomedical applications. On account of surface chemistry, its affinity to bind with drugs, receptors, ligands, etc. is very high [7]. Natural biopolymers such as pectin, alginate and chitosan have recently found wide applications in food and pharmaceutical sectors [8]. The presence of functional groups in the biopolymer structures not only help to encapsulate bioactive compounds or drugs but also to form a cross-linked dense network after encapsulation [9].

Recently pectin-based delivery systems have been registered for colon-specific delivery purposes. Pectin is a natural, anionic polysaccharide obtained from the cell walls of most fruits, e.g. apples, oranges and pears [10]. Though it is resistant to the enzymes present in upper gastro-intestinal tract, it undergoes complete degradation by colonic bacterial enzymes [11]. Chitosan is another linear polysaccharide with cationic properties which is obtained from chitin, a major component of the shells of shrimp, lobsters and crabs. Pectin and chitosan are known to combine to form a polyelectrolyte complex which can be used to encapsulate bioactive molecules and drugs [12].

Resveratrol is a polyphenolic compound found in red grapes, cranberries and blueberries that possesses a wide range of beneficial effects on human health [13,14]. It is found in two different structural configurations, cis-resveratrol and trans-resveratrol. Trans-configuration is the dominant configuration which represents the most thoroughly studied chemical form [15]. Recently, considerable attention has been paid to the molecule as several health benefits of resveratrol, such as anti-inflammatory activity, hepatoprotection, cardio- and neuro-protection, anti-cancer activity, anti-ageing effects and diabetes prevention [16,17] are reported. Despite the multiple health benefits of this natural molecule, it is often difficult to incorporate resveratrol into commercial pharmaceutical products because its poor water solubility and chemical instability lowers the bioavailability [18,19]. Moreover, free resveratrol gets metabolized quite rapidly in biological systems.

The benefits of resveratrol are only realizable when it is delivered in an encapsulated form in the body. However, this approach has not been fully explored and only a few studies have been reported in this area. Ribeiro et al. have reported the preparation of biohybrid beads coated with pectin and studied colon-specific drug delivery taking 5-aminosalicylic acid as model drug [20]. Rebitski et al. [21] have also prepared the core–shell beads with clay as host into chitosan/pectin core and was used in the delivery of metformin. Our work, however, focuses mainly on a specific drug, resveratrol which, in spite of its strong antioxidant and cancer-preventing activity, is not effective as an oral drug due to poor bioavailability and rapid metabolism.

Therefore, the aim of the present study is to develop chitosan–pectin core–shell nanoparticles loaded with resveratrol in order to enhance its bioavailability and then analyse the effect of thickness of the pectin-shell on the drug release efficiency of the formulation. It is also intended to investigate the effect of encapsulation on the radical scavenging activity of resveratrol.

# 2. Material and methods

Chitosan (from shrimp shells, molecular weight—3800–20 000 Dalton) and pectin (galacturonic acid greater than or equal to 65.0%) were purchased from HIMEDIA. Resveratrol (greater than 99.0%) which is chemically 3,5,4′-trihydroxy-trans-stilbene was procured from Tokyo Chemical Industry, Japan (TCI). 2,2-Diphenyl-1-picrylhydrazyl free radical (DPPH) was purchased from Alfa Aesar. L-ascorbic acid (greater than 99%) was obtained from Fisher Chemical. Buffer capsules (pH 4, 7, 9) were purchased from MERCK and all other chemicals and solvents used such as ethanol, methanol, hydrochloric acid and acetic acid were of analytical grade.

Resveratrol-encapsulated chitosan–pectin core–shell nanoparticles (RCP1) were synthesized using antisolvent precipitation and electrostatic deposition methods [22]. Chitosan was loaded with resveratrol using sonication method in order to fabricate the nano-core. Pectin was chosen as shell material. The thickness of the shell was varied by changing the proportion of pectin (20 to 70 ml) in three experiments (RCP1, RCP2 and RCP3) keeping the same loading of resveratrol in chitosan-core.

All the samples were characterized by Fourier transform infrared spectrometer (FT-IR) spectroscopic method. For sample preparation, KBr pellet method was used. The sample was mixed with anhydrous potassium bromide in a ratio of 1 : 100 and ground to a fine powder in a mortar-pestle. The mixture was transferred into pellet press dies and pressed in a manual hydraulic pellet press machine to obtain a pellet of the sample mixed with KBr. The pellet was then placed in ALPHA Bruker FT-IR spectrometer in order to record the spectrum over a wavenumber range of 4000–500 cm$^{-1}$.

RCP1 was characterized with the help of field-emission scanning electron microscope (FESEM) of model Zeiss Sigma 300 in order to examine the external morphology of the synthesized core–shell nanoparticle. The core–shell nanoparticles RCP1 were characterized by field-emission transmission electron microscope/selected area electron diffraction (FETEM/SAED) of model JEOL, 2100F.

RCP2 and RCP3 were characterized by high-resolution transmission electron microscope (HRTEM) of model JEOL, JEE 2100 in order to observe the internal morphology of the nanoparticles.

The hydrodynamic diameters of the resveratrol-loaded chitosan–pectin core–shell nanoparticles were measured using zeta sizer Nano ZS90 instrument (model no. ZEN3690). For the sample preparation in dynamic light scattering (DLS) experiment, 0.025 g each of RCP1, RCP2 and RCP3 was added to deionized water and sonicated for 2 h. The dispersion was then used for DLS measurements.

The release efficiency of resveratrol from all three resveratrol-encapsulated chitosan–pectin nanoparticles was then studied at various pH for a certain period of time using the UV-Visible spectrophotometer.

The DPPH scavenging activity of encapsulated resveratrol and free resveratrol (RSV) was determined using a method reported previously by Huang et al. [13], with the help of a UV spectrophotometer taking L-ascorbic acid as a standard.

## 2.1. Experimental

### 2.1.1. Loading of resveratrol in chitosan

A weighed mass of 0.5 g chitosan was dissolved in 100 ml of 85% (v/v) aqueous ethanol solution (as chitosan is difficult to be dissolved, few drops of acetic acid were added while preparing its solution) with continuous magnetic stirring at 500 r.p.m. for 3 h. Then 0.1 g of powdered resveratrol, dissolved in 15 ml of ethanol solution was added to the chitosan solution. The resulting mixture from this antisolvent precipitation method was sonicated for 1 h in order to obtain resveratrol-loaded chitosan nanoparticles.

### 2.1.2. Preparation of pectin solution

A 0.5 g of powdered pectin was dissolved in distilled water and heated at 70°C with continuous stirring for 30 min. The solution was then cooled to ambient temperature and stirred again for 1 h. The pH of the solution was adjusted to 4.0 by adding 0.1 M HCl dropwise.

### 2.1.3. Preparation of resveratrol-loaded chitosan–pectin core–shell nanoparticles

Resveratrol-loaded chitosan–pectin core–shell nanoparticles were prepared using antisolvent precipitation and electrostatic deposition method with slight modification as reported previously [22]. Five millilitres of ethanolic suspension of resveratrol-loaded chitosan was rapidly injected into 20 ml water (adjusted to pH 4.0) with the help of a syringe. The mixture was then continuously stirred at 900 r.p.m. using a magnetic stirrer. This process led to the formation of a suspension of resveratrol-loaded chitosan nanoparticles. The resulting dispersion was stirred for another 3 min, and ethanol was evaporated from the nanoparticle suspension by heating the solution in a water bath at 80°C for 15 min. The appropriate amount of water (adjusted to pH 4.0) was added to the suspension to compensate for any decrease in volume due to the evaporation of ethanol. The dispersion was poured into 35 ml pectin solution in water with continuous stirring for 30 min. The resulting suspension was then centrifuged at 3000 r.p.m. for 30 min, and the settled residue was dried in a vacuum desiccator for 24 h in order to obtain the desired resveratrol-loaded chitosan–pectin core–shell nanoparticles (RCP1).

**Scheme 1.** Fabrication of resveratrol loaded chitosan–pectin core–shell nanoparticles.

The same procedure was followed in the preparation of two other core–shell nanoparticles RCP2 (with higher proportion of pectin compared with RCP1, i.e. 75 ml) and RCP3 (with less proportion of pectin compared with RCP1, i.e. 20 ml), in which the shell thickness was varied by varying the proportion of pectin but keeping the amount of resveratrol-loaded chitosan same. The synthesis of core–shell nanoparticles is given in scheme 1.

## 2.2. Particle yield and resveratrol loading efficiency

Freshly prepared colloidal dispersions of the synthesized core–shell nanoparticles RCP1 were centrifuged at 1000 r.p.m. for 10 min in order to separate any large particles and then the core–shell nanoparticles in the serum phase were dried in a vacuum desiccator and weighed. The amount of resveratrol loaded in the prepared core–shell nanoparticles was determined using UV-visible spectrophotometer. For this, the sample was dissolved in ethanol and the concentration of resveratrol was determined using the calibration graph.

The particle yield and the resveratrol loading efficiency were calculated using the following equations:

$$\text{particle yield } (\%) = \frac{\text{the weight of the freeze dried particles}}{\text{total weight of resveratrol; chitosan pectin}} \times 100\%$$

$$\text{loading efficiency } (\%) = \frac{\text{resveratrol} \in \text{nanoparticles}}{\text{total resveratrol input}} \times 100\%.$$

The particle yield (P) was found to be 89%, and the percentage of drug (resveratrol) loading (L) was found to be 55%, which is considered good according to the literature [23].

## 2.3. Study of drug (resveratrol) delivery from RCP1, RCP2 and RCP3

The *in vitro* drug release from the prepared core–shell nanoparticles was monitored with the help of UV-Visible spectrophotometer. A 0.1 g sample of RCP1 was added to 100 ml of distilled water in a beaker. The spectrum was recorded and the absorbance was checked at 285 nm. The solutions of different pH were prepared using buffer capsules of three different pH values (4, 7 and 9).

## 2.4. Evaluation of antioxidant activity of RCP1 (chitosan-encapsulated resveratrol) using DPPH radical

To determine the DPPH scavenging activity of the sample, methanolic suspension of RCP1 (with a shell of pectin around) and free resveratrol were prepared at multiple concentrations. For each antioxidant, four different concentrations were tested. First of all, 100 µM DPPH solution was prepared in methanol. To obtain the absorbance of the control reaction, the DPPH solution was diluted 10 times,

and its absorbance was measured at wavelength 518 nm. Then for encapsulated resveratrol and free resveratrol, four different concentrations, 30, 50, 80 and 100 µg ml$^{-1}$, were prepared considering the loading efficiency of encapsulated resveratrol. After that 2 ml of 100 µM DPPH solution was mixed with equivalent aliquots of RCP1 and free resveratrol. It was then incubated for 30 min in the dark at room temperature. After that, the absorbance of the reaction mixture was measured at wavelength 518 nm. L-ascorbic acid was taken as a standard and its antioxidant activity was measured for comparison using the same process. The percentage of DPPH radical scavenging activity was calculated by using the following equation:

$$\text{DPPH radical scavenging (\%)} = \frac{A_c - A_s}{A_c} \times 100\%,$$

where $A_c$ is the absorbance of the control reaction and $A_s$ is the absorbance of the test sample.

# 3. Results and discussions

## 3.1. Characterization of resveratrol-loaded chitosan–pectin core–shell nanoparticles

### 3.1.1. Fourier transform infrared spectrometer study

The FT-IR spectra of chitosan (figure 1*a*) show a broad band at 3451 cm$^{-1}$ for N-H symmetrical stretch overlapped with O-H stretching band. The band at 2925 cm$^{-1}$ is due to C-H stretching. The chitosan polymer shows characteristic band of amide I (C=O stretch) at 1637 cm$^{-1}$ and amide II (C-N stretching and N-H bending) at 1542 cm$^{-1}$. The band for amide III (N-H deformation vibration coupled with C-N stretching) is observed at 1374 cm$^{-1}$ [24]. Another band at 1160 cm$^{-1}$ is due to C-O-C stretching of the saccharide units. The band at 1050 cm$^{-1}$ is for NH bending vibration.

The FT-IR spectra of pectin (figure 1*b*) show a broad band at 3410 cm$^{-1}$ due to O-H stretching. The band at 2925 cm$^{-1}$ is due to C-H stretching. The polymer shows a typical band at 1754 cm$^{-1}$ due to the presence of ester carbonyl C=O group [25]. Bands at 1644 and 1408 cm$^{-1}$ correspond to the COO- group stretching (range 1600–1650 cm$^{-1}$ for antisymmetric stretch and 1400–1450 cm$^{-1}$ for symmetric stretch). Another band at 1102 cm$^{-1}$ is observed due to the presence of ether linkage and glycosidic bond in pectin. An absorption band at 1060 cm$^{-1}$ is observed due to C-H bending.

The FT-IR spectra of resveratrol (figure 1*c*) show a narrow band at 3298 cm$^{-1}$ for phenolic O-H stretching. There are several intense bands in 1650 to 1000 cm$^{-1}$ region out of which the band at 1383 cm$^{-1}$ is due to C–O stretching of a phenolic group. The band at 1586 cm$^{-1}$ corresponds to C=C olefinic stretching and the band at 1606 cm$^{-1}$ is due to C=C aromatic double-bond stretching. The peak at 965 cm$^{-1}$ is the characteristic of trans-olefinic stretch which confirms the presence of resveratrol in trans-isomer form. The band at 831 cm$^{-1}$ is the characteristic of an olefinic group [26,27].

The FT-IR spectra of RCP1 (figure 1*d*) exhibit three strong absorption bands at 1637, 1094 and 1340 cm$^{-1}$ due to C–C aromatic double-bond stretching, C–C olefinic stretching, and C–C stretching, respectively, in resveratrol [28,29]. The peak at 3450 cm$^{-1}$ has been attributed to the vibrational stretching of O-H bond of resveratrol-loaded chitosan–pectin core–shell nanoparticles. Peak at 685 cm$^{-1}$ corresponds to a C-H stretch from the aromatic ring. The comparison of FT-IR spectrum of RCP1 with that of free chitosan, pectin and resveratrol reveals that there is no major functional change in the components after encapsulation. However, the band at 1542 cm$^{-1}$ in chitosan for amide II stretch was very much reduced in intensity along with the band at 1644 cm$^{-1}$ for carboxyl group stretching in pectin. The intensity of the band at 1754 cm$^{-1}$ due to methyl ester of pectin is also reduced to a large extent indicating the intermolecular electrostatic interaction of carboxyl group of pectin with the amino group of chitosan.

The FT-IR spectra of RCP2 and RCP3 were also compared with the FT-IR spectra of RCP1 (electronic supplementary material, figure S4). No difference was found in the three spectra which indicate that the interaction between the polymers is not changed on varying the ratio of chitosan and pectin in the prepared core–shell nanoparticles.

### 3.1.2. Field-emission scanning electron microscopy

FESEM images of RCP1 (figure 2) were recorded at different (100 and 30 nm) scales in order to determine the external morphology of the synthesized core–shell nanoparticles (with chitosan-loaded resveratrol as core and pectin as shell). The images revealed the formation of spherical-shaped core–shell nanoparticles

**Figure 1.** FT-IR spectra of (*a*) free chitosan, (*b*) pectin, (*c*) resveratrol and (*d*) RCP1.

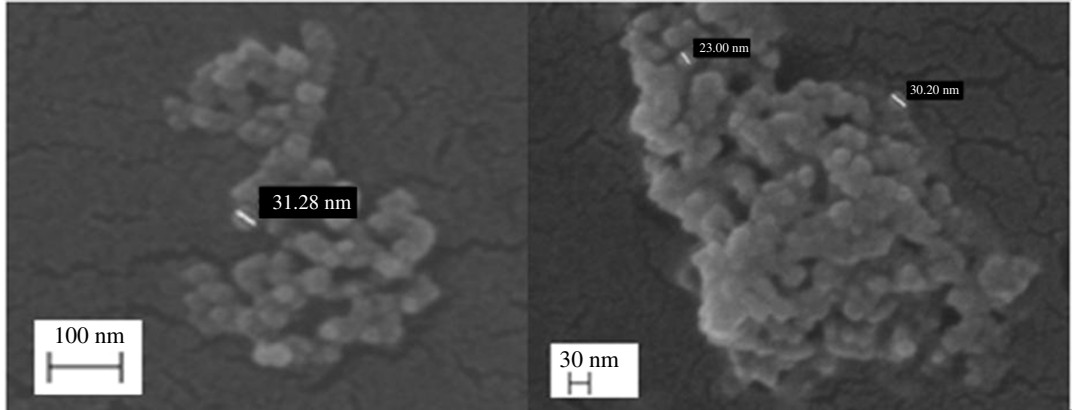

**Figure 2.** FESEM images of RCP1.

(electronic supplementary material, figure S1). The sizes of spherical-shaped core–shell nanoparticles are in the range 23–32 nm.

### 3.1.3. Field-emission transmission electron microscopy/selected area electron diffraction

FETEM/SAED images (figure 3*a*) were recorded for RCP1 in order to find out the internal morphology and distribution of particles in the core–shell nano-drug carrier. The images recorded at different scales (100, 50, 20 and 5 nm) clearly revealed the formation of monodispersed core–shell nanoparticles with resveratrol-loaded chitosan as the core and pectin as shell. The FETEM images were manually

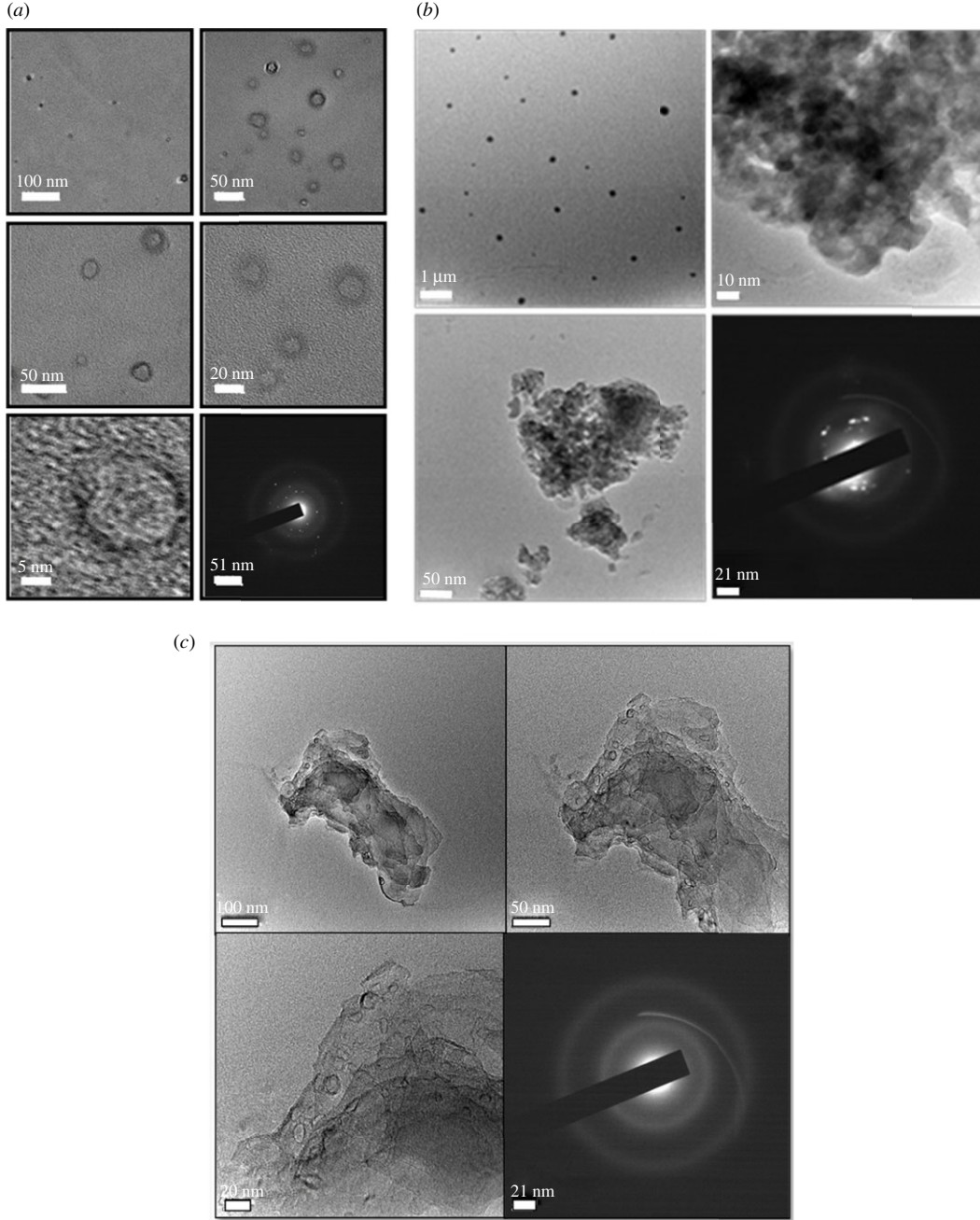

**Figure 3.** (*a*) FETEM/SAED images of RCP1. (*b*) HRTEM/SAED images of RCP2. (*c*) HRTEM/SAED images of RCP3.

analysed, and it was found that the average diameter of the spherical core (prepared of chitosan-uploaded resveratrol) is approximately 13 nm and the average thickness of the shell is roughly 4 nm. The average diameter of each core–shell nanoparticle appeared to be 21 nm. The SAED pattern recorded at 51 nm scale showed some bright circles with bright spots confirming highly crystalline behaviour of the resveratrol loaded in chitosan.

HRTEM images of RCP2 revealed the increase in thickness of the pectin-shell around resveratrol-loaded chitosan-core with increasing proportion of pectin during synthesis of the core–shell nanoparticles. However, some degree of agglomeration was witnessed in the TEM images recorded at various scales at this relative concentration of the pectin and resveratrol-loaded chitosan (figure 3*b*). HRTEM images of RCP3 clearly showed the formation of resveratrol-loaded chitosan–pectin core–shell nanoparticles with shell thickness ranging in between 2–3 nm (figure 3*c*). The observed crystalline nature of the as-synthesized core–shell nanoparticles was further confirmed by the lattice planes visible in high-resolution TEM image recorded taking scale of 5 nm (electronic supplementary material, figures S2 and S3).

**Table 1.** Variation of shell thickness in RCP1, RCP2 and RCP3 as revealed from TEM images.

| resveratrol loaded chitosan : pectin | RCP1 1 : 7 | RCP2 1 : 15 | RCP3 1 : 4 |
|---|---|---|---|
| TEM images at 50 nm scale |  |  |  |
| shell thickness | 4.00 nm | 6.25 nm | 2.50 nm |
| average diameter of each core–shell nanoparticle | 21.00 nm | 21.80 nm | 10.93 nm |

The extent of pectin coating over the surface of resveratrol-loaded chitosan nanoparticles was determined from HRTEM images of each RCP1, RCP2 and RCP3. Several images of each sample with wide distribution of core–shell nanoparticles were selected. The size of several core–shell nanoparticles in a particular image was measured using the scale bar provided in that image. From the contrast difference of chitosan and pectin, the shell thickness was measured. The average value of the thickness of pectin of 20 to 30 core–shell nanoparticles of each sample was considered as final value. The overall results of the analysis of TEM images of RCP1, RCP2 and RCP3 are presented in table 1.

### 3.1.4. Dynamic light scattering experiment

The DLS experiments show the size distribution of the particles of RCP1, RCP2 and RCP3 with respect to intensity (figure 4a–c). The sizes (hydrodynamic diameter values) calculated from DLS studies were much higher than the sizes reported from TEM studies. This is not unusual because the TEM image gives the actual size of the particles determined on a statistically small sample whereas DLS provides the hydrodynamic diameter which is the diameter of a hypothetical sphere having a similar diffusion coefficient in the same environment of the particles. When the nanoparticle is dispersed, its surface shows strong interactions with solvent molecules, which strongly influences behaviour of the nanoparticles. The presence of surface charges on the particles causes repulsion between the particles and helps to prevent agglomeration.

### 3.1.5. Zeta potential measurement

Nanoparticles have a surface charge that attracts a thin layer of ions of opposite charge to the nanoparticle surface. This double layer of ions travels with the nanoparticle as it diffuses throughout the solution. The electric potential at the boundary of the double layer is known as the zeta potential of the particles. The zeta potential, which depends on the surface charge, is an important factor for the stability of nanoparticles in suspension and is also the major factor in the initial adsorption of nanoparticles onto the cell membrane. The values of zeta potential can range from +100 to −100 mV. Nanoparticles with zeta potential values greater than +25 mV or less than −25 mV typically have high degrees of stability. Dispersions with a low zeta potential value will eventually aggregate due to Van der Waal inter-particle attractions [30,31].

Zeta potential measurements of the three samples RCP1, RCP2 and RCP3 show the magnitude of the surface charge of the colloidal particles of each sample, as given in table 2.

The magnitude of the surface charge was found to be higher in the case of RCP2, which may be due to the thicker pectin-shell with polar end groups. In the case of RCP1 and RCP3, due to interaction between the two biopolymers (pectin and chitosan), the charges are somewhat diminished depending upon the ratio of the two. This is why the zeta potential values are found to be low. The zeta potential value of the RCP2 having chitosan : pectin in the ratio 1 : 15 indicated good stability [32], while the values of RCP1 and RCP3 indicate lower stability when compared with the other two. However, none of the values are suggestive of rapid agglomeration.

Although the dilute samples used for DLS and zeta potential measurements cannot adequately represent the therapeutic formulations to be used *in vivo*, they primarily indicate the potential of the use of the formula to design a drug delivery vehicle.

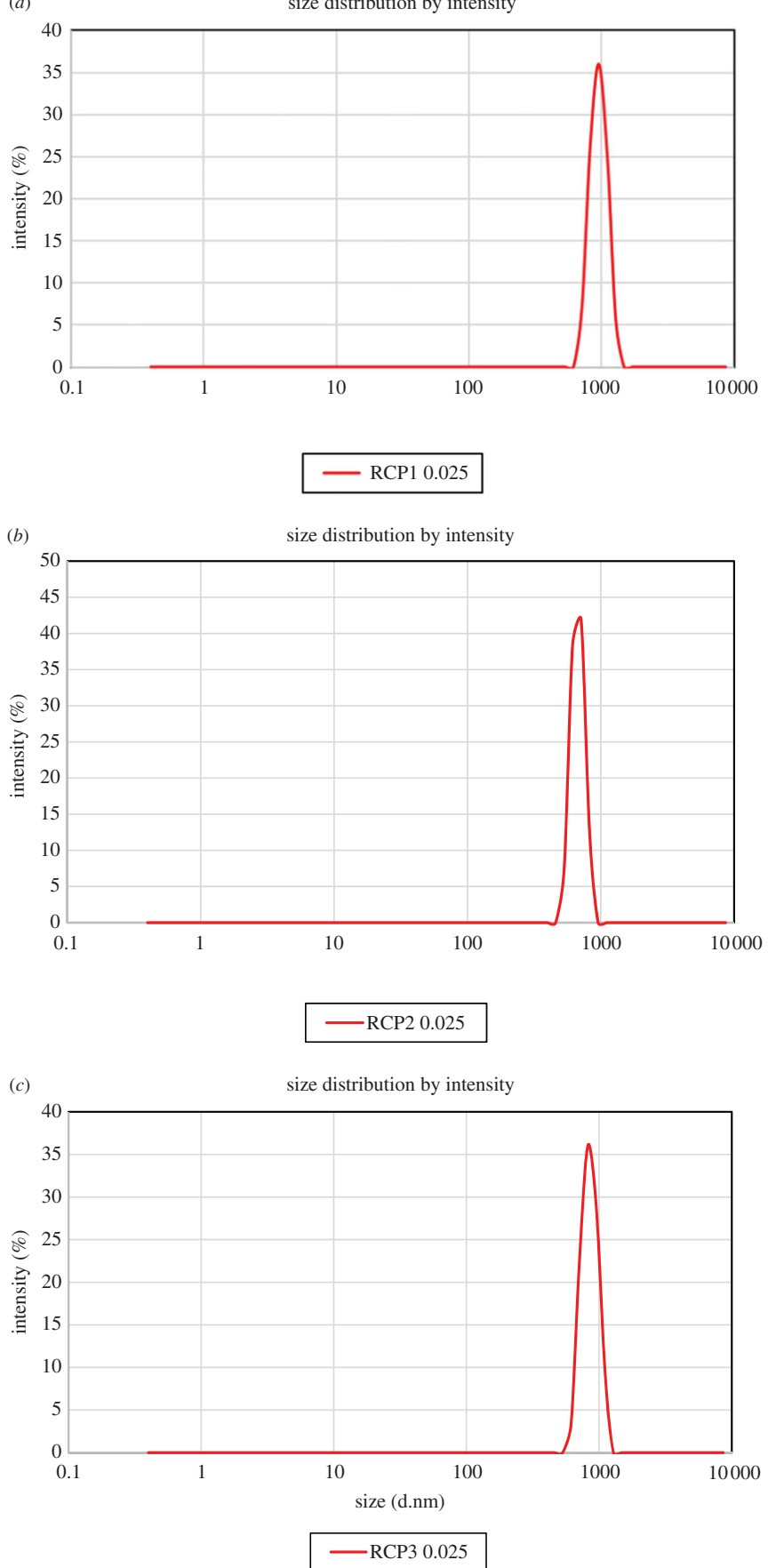

**Figure 4.** (*a*) Hydrodynamic diameter of RCP1 measured using DLS. (*b*) Hydrodynamic diameter of RCP2 measured using DLS. (*c*) Hydrodynamic diameter of RCP3 measured using DLS.

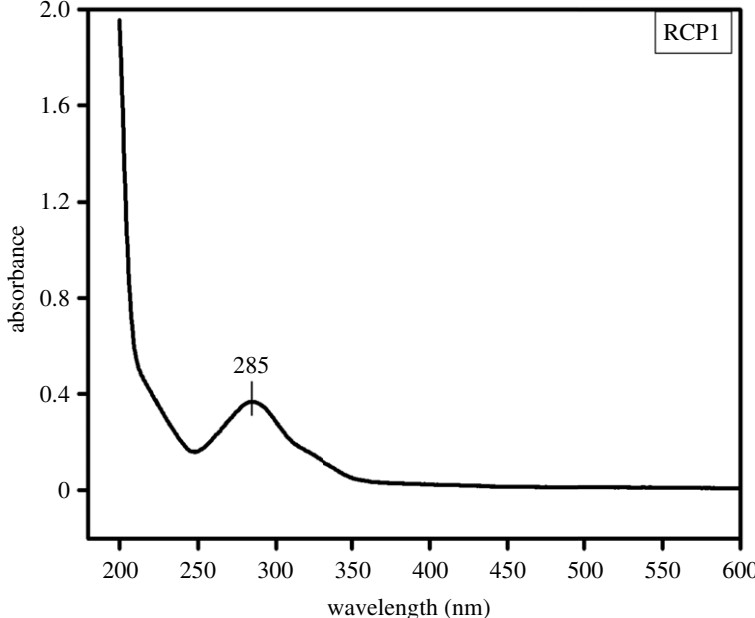

**Figure 5.** UV-Vis spectra of resveratrol released from RCP1.

**Table 2.** Zeta potential of the colloidal particles of RCP1, RCP2 and RCP3.

| resveratrol-loaded chitosan–pectin core–shell nanoparticles | RCP1 | RCP2 | RCP3 |
|---|---|---|---|
| chitosan : pectin | 1 : 7 | 1 : 15 | 1 : 4 |
| zeta potential at a concentration of 0.025% | −12.2 mV | −47.7 mV | + 10.8 mV |

## 3.2. Effect of pH on release efficiency of the drug from core–shell nanoparticles

Trans-isomer of resveratrol was used in the samples of core–shell nanoparticles prepared for drug delivery. But on UV-irradiation, the trans-isomer is converted into cis-isomer [33,34]. Figure 5 shows the absorbance peak at 285 nm due to cis-resveratrol released by RCP1. However, the peak due to trans-resveratrol was obtained as only a low intense shoulder in between 300 and 350 nm. This is why the absorbance was checked at 285 nm and not at around 304 nm for the *in vitro* drug release from the prepared core–shell nanoparticles.

For a nano-drug to exhibit a beneficial health effect, it is important that the nutraceuticals get released in the gastro-intestinal tract, GIT [35]. For this very reason, the impact of pH was monitored on the drug release efficiency of the synthesized core–shell nanoparticles. Three different pH values (4, 7 and 9) were chosen. Interestingly, a sustained release of the drug from RCP1 was observed at both acidic and alkaline pH. The release was much less at the neutral pH (figure 6a). Since our stomach has gastric juice with acidic pH and the digestive juice in the intestinal tract has alkaline pH, the formulation can show sustained release not only in acidic pH of the stomach but also in the alkaline pH of the intestine. This feature of the prepared core–shell nanoparticles is unique as it targets different organ sites with more efficacies for various health benefits.

## 3.3. Effect of shell thickness on the release efficiency of the drug

The drug release efficiency was checked for both thick- (RCP2) and thin-shelled (RCP3) core–shell nanoparticles and was compared with that of RCP1. Both RCP2 and RCP3 showed faster drug release in acidic and alkaline pH compared with the neutral medium as was observed with RCP1. However, the three samples were different in the percentage of drug delivered with time. In the case of RCP1, less than 60% of the drug was released in the first 24 h and the sustained release behaviour continued even up to 30 h indicating that the drug is released from the matrix slowly. Moreover, the release of the drug from RCP1 followed the similar kinetics at all the pH ranges studied. In the case of RCP2

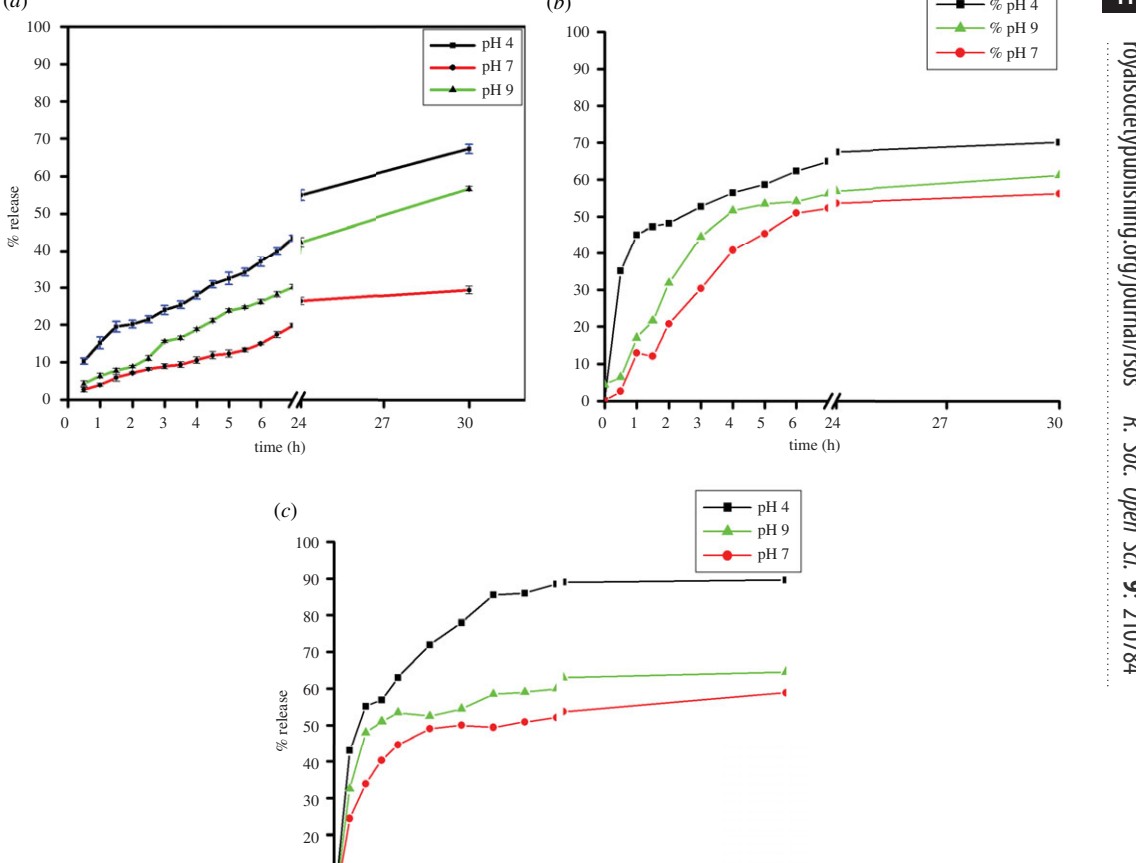

**Figure 6.** (*a*) Time versus % release graph of resveratrol from RCP1 at various pH. (*b*) Time versus % release graph of resveratrol from RCP2 at various pH. (*c*) Time versus % release graph of resveratrol from RCP3 at various pH.

(figure 6*b*), about 45% of the drug was released within first 1 h in acidic medium but the rate of release became slower subsequently. The release pattern was found to be similar at other pH values of the medium. About 60% of the drug was released within 24 h and then the release almost stopped. RCP3, however, showed more than 80% release of the drug within first 4 h in acidic medium and 90% of the drug was released in 30 h (figure 6*c*). In alkaline and neutral medium, about 50% of the drug was released in first 4 h and continued up to 60% in 30 h showing better release profile.

Therefore, among the three samples RCP1, RCP2 and RCP3 having chitosan and pectin ratio 1 : 7, 1 : 15 and 1 : 4, respectively, RCP1 with medium pectin-shell thickness was the best drug carrier with sustained release efficiency. The ratio of the two polysaccharides plays a significant role in the stability of the prepared samples and the kinetics of the drug release. When the ratio of the two polysaccharides, chitosan and pectin, is changed, the rate of drug release decreases with the increasing thickness of pectin-shell. However, the sustained release was observed when the ratio is 1 : 7. Therefore, the thickness of pectin-shell in these core–shell nanoparticles must be optimized taking into consideration the sustained release of drug from the newly designed drug delivery vehicle.

Pectin derivatives carry a high charge density (either positive primary amine or negative carboxyl groups) and are able to penetrate deeply into tissue. Formulations with pectin derivatives show a better ability to retain the incorporated drugs. The pectin coating not only significantly increased the capability of chitosan nanoparticles to encapsulate resveratrol but also influenced the rate of release. Therefore, pectin coating can serve as an effective strategy to stabilize chitosan-loaded drug nanoparticles as oral delivery vehicles.

The core–shell nanoparticles designed in this way are effective carriers for resveratrol. As nanoparticles can better penetrate the tumour cells, the prepared compositions can be more effective in preventing and curing cancer.

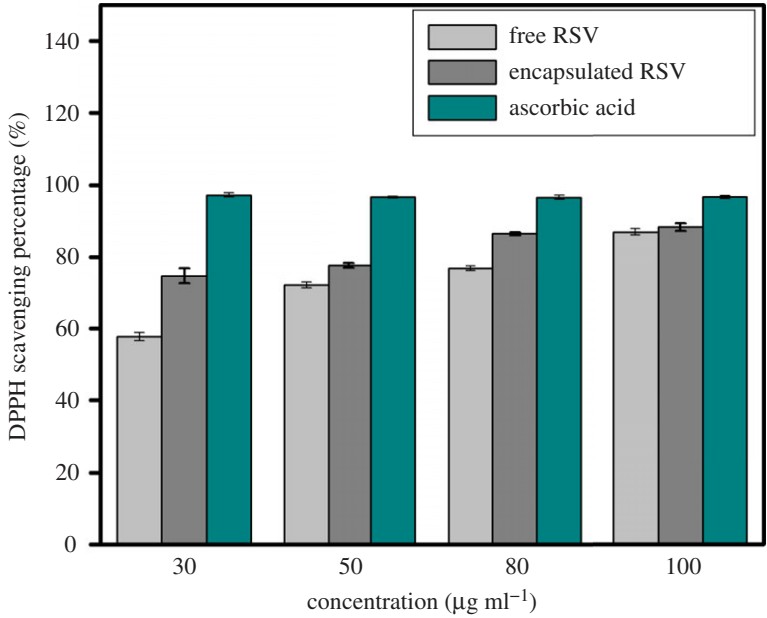

**Figure 7.** DPPH radical scavenging percentage of ascorbic acid, free RSV and encapsulated RSV, RCP1 (in methanol) at different concentrations.

## 3.4. Antioxidant activity of encapsulated resveratrol

Resveratrol is known to be an excellent scavenger of hydroxyl, superoxide and many other radicals [36]. It is highly soluble in methanol. The DPPH scavenging activities of encapsulated resveratrol and free resveratrol were compared with standard L-ascorbic acid. The DPPH scavenging percentage increases linearly with increasing ascorbic acid concentration. The DPPH scavenging percentage of both encapsulated and free resveratrol also increases with increasing concentration of resveratrol from 30 to 100 µg ml$^{-1}$. However, encapsulated resveratrol is found to have a significantly higher scavenging activity than free resveratrol (figure 7). Mariarenata *et al.* [37] has also reported that encapsulated resveratrol exhibit higher chemical and cellular antioxidant activities than free resveratrol. Statistical analysis using two-way ANOVA in MS Excel indicates from the comparison of F values with F critical values and P values with $\alpha$ values that encapsulated resveratrol is significantly different and better than free RSV in antioxidant activity (electronic supplementary material, table S1).

## 4. Conclusion

Resveratrol-loaded chitosan–pectin core–shell nanoparticles were successfully fabricated by encapsulating resveratrol in cationic chitosan and coating it with anionic pectin-shell through antisolvent precipitation and electrostatic deposition methods. Resveratrol is not freely soluble in water and is only slightly soluble in oil. Therefore, colloidal delivery systems which use oil as a carrier phase, such as nano-emulsions or solid lipid nanoparticles show limited performance due to their low loading efficiency. However, the synthesized core–shell nanoparticles were able to encapsulate an appreciable amount of resveratrol at a loading efficiency greater than 55% with particle yield of 89%.

Another major finding of this research work is that encapsulation of resveratrol into biopolymer core–shell nanoparticles significantly enhances its bioavailability. The rapid metabolism of resveratrol seems to be a limiting factor in translating its promising health benefits in humans. In this research work, we have tried to enhance the bioavailability of resveratrol by encapsulating it in a core–shell biopolymer nanoparticle. This eventually leads to its seamless delivery in the body, through sustained way and not in a burst. Interestingly a sustained release pattern of resveratrol for almost 30 h was obtained in the present research work. The release efficiency is dependent on both pH and shell thickness. The fraction of resveratrol released from the nanoparticles in a sustained way was more in both acidic and alkaline pH compared with the neutral pH. Therefore, the sustained release for as long as 30 h may be obtained not only in acidic pH of the stomach but also in the alkaline pH of the intestine. This unique feature of the prepared core–shell nanoparticles targets it to different organs with better efficacy.

Additionally, drug release kinetics can be controlled by varying the thickness of the pectin-shell. The thinner shell promotes a better release of the drug. The encapsulated resveratrol which is more stable in the form of core–shell nanoparticles also exhibited stronger more and potent *in vitro* antioxidant activity than the free resveratrol due to increased surface area.

Finally, our study suggests a simple but novel method for the development of core–shell nanoparticles using two biocompatible polymers as a drug delivery vehicle to facilitate sustained release of the loaded drug. Therefore, the core–shell nanoparticles fabricated in this research work may be used as a potential drug delivery vehicle in future.

Ethics. This statement is not relevant to our work.

Data accessibility. The datasets supporting this article are available at: https://doi.org/10.5061/dryad.41ns1rndk. The data are provided in the electronic supplementary material [38].

Authors' contributions. S.S.: data curation, investigation, methodology, visualization, writing—original draft and writing—review and editing; S.A.: conceptualization, data curation, methodology, supervision, visualization, writing—original draft and writing—review and editing; P.B.: conceptualization, data curation, investigation, methodology, software and validation; J.H.: conceptualization, formal analysis, investigation, methodology, software and visualization; M.G.: conceptualization, data curation, formal analysis, investigation, methodology, project administration, resources, supervision, writing—original draft and writing—review and editing

All authors gave final approval for publication and agreed to be held accountable for the work performed therein.

Competing interests. We have no competing interests.

Funding. No source of funding for any author.

Acknowledgements. The authors would like to thank SAIF, NEHU (Shillong) for providing HRTEM/SAED instrument facility; IIT Guwahati for FETEM/SAED, DLS and zeta potential measurement and Gauhati University for FESEM instrument facilities

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
