## [Peer Review File · Royal Society Open Science]

Review History

RSOS-210784.R0 (Original submission)

Review form: Reviewer 1

Is the manuscript scientifically sound in its present form?

Yes

Are the interpretations and conclusions justified by the results?

No

Is the language acceptable?

Yes

Do you have any ethical concerns with this paper?

No

Have you any concerns about statistical analyses in this paper?

No

Recommendation?

Accept with minor revision (please list in comments)

Comments to the Author(s)

In this article, the authors report on nanoparticles of Resveratrol Loaded Chitosan-Pectin Core-Shell as a New Drug Delivery Vehicle for Continuous Release and Enhanced Antioxidant Activity. The author has conveyed in detail and complete data.

Recommendation: Minor revisions

1. The in vitro drug release from the prepared core-shell nanoparticles was monitored with the help of a UV-Visible spectrophotometer in the absorbance was checked at 285 nm. Does this mean that the resveratrol used is cis-resveratrol? The material and method only convey the purity and origin of the resveratrol used. While in schematic A you write down the structure of the compound of trans-resveratrol. The value of the maximum lambda of trans-resveratrol, not 285 nm.
2. The length of the introduction should be reduced. It is too lengthy. It would be better if only the important points were written down and some were transferred to the results and discussion.

Review form: Reviewer 2

Is the manuscript scientifically sound in its present form?

Yes

Are the interpretations and conclusions justified by the results?

Yes

Is the language acceptable?

Yes

Do you have any ethical concerns with this paper?

No

Have you any concerns about statistical analyses in this paper?

Yes

Recommendation?

Major revision is needed (please make suggestions in comments)

Comments to the Author(s)

File attached (see Appendix A).

Decision letter (RSOS-210784.R0)

Dear Dr Ganguly:

Title: Resveratrol Loaded Chitosan-Pectin Core-Shell Nanoparticles as Novel Drug Delivery Vehicle for Sustained Release and Improved Antioxidant Activities
Manuscript ID: RSOS-210784

The editor assigned to your manuscript has now received comments from reviewers. We would like you to revise your paper in accordance with the referee and Subject Editor suggestions which can be found below (not including confidential reports to the Editor). Please note this decision does not guarantee eventual acceptance.

Please submit your revised paper before 17-Sep-2021. Please note that the revision deadline will expire at 00.00am on this date. If we do not hear from you within this time then it will be assumed that the paper has been withdrawn. In exceptional circumstances, extensions may be possible if agreed with the Editorial Office in advance. We do not allow multiple rounds of revision so we urge you to make every effort to fully address all of the comments at this stage. If deemed necessary by the Editors, your manuscript will be sent back to one or more of the original reviewers for assessment. If the original reviewers are not available we may invite new reviewers.

Yours sincerely,
Dr Ellis Wilde
Publishing Editor, Journals

On behalf of the Subject Editor Professor Anthony Stace and the Associate Editor Professor Chaohua Cui.

RSC Associate Editor
Comments to the Author:
(There are no comments.)

RSC Subject Editor
Comments to the Author:
(There are no comments.)

Reviewers' Comments to Author:

Reviewer: 1

Comments to the Author(s)

In this article, the authors report on nanoparticles of Resveratrol Loaded Chitosan-Pectin Core-Shell as a New Drug Delivery Vehicle for Continuous Release and Enhanced Antioxidant Activity. The author has conveyed in detail and complete data.

Recommendation: Minor revisions

1. The in vitro drug release from the prepared core-shell nanoparticles was monitored with the help of a UV-Visible spectrophotometer in the absorbance was checked at 285 nm. Does this mean that the resveratrol used is cis-resveratrol? The material and method only convey the purity and origin of the resveratrol used. While in schematic A you write down the structure of the compound of trans-resveratrol. The value of the maximum lambda of trans-resveratrol, not 285 nm.
2. The length of the introduction should be reduced. It is too lengthy. It would be better if only the important points were written down and some were transferred to the results and discussion.

Reviewer: 2

Comments to the Author(s)

File attached

Author's Response to Decision Letter for (RSOS-210784.R0)

See Appendix B.

RSOS-210784.R1 (Revision)

Review form: Reviewer 3

Is the manuscript scientifically sound in its present form?

No

Are the interpretations and conclusions justified by the results?

Yes

Is the language acceptable?

Yes

Do you have any ethical concerns with this paper?

No

Have you any concerns about statistical analyses in this paper?

No

Recommendation?

Accept with minor revision (please list in comments)

Comments to the Author(s)

All comments were viewed and responded to by authors but the quality of figures is not good and they must be improved.

Review form: Reviewer 4

Is the manuscript scientifically sound in its present form?

No

Are the interpretations and conclusions justified by the results?

No

Is the language acceptable?

Yes

Do you have any ethical concerns with this paper?

No

Have you any concerns about statistical analyses in this paper?

No

Recommendation?

Reject

Comments to the Author(s)

This manuscript introduces the development of core-shell nanoparticles to be used in the controlled release of resveratrol. The topic could be interesting however, there are significant drawbacks in the reported study that precludes its recommendation for being published. I will signal some of the most significant ones with the aim help authors to improve their work before intend its publication:

The study seems to address the preparation of nanoparticles formed by a core of chitosan and an external coating of pectin to be used as a controlled drug delivery for oral administration.

- Firstly, I have various issues that arise me doubts about the goal, adequate situation of the study in relation to previous ones and the validation of the study with respect the final goal:

1) There are reports in the literature addressing the fabrication of core-shell structures of similar structure for oral drug administration, from the report by Ribeiro et al. "Pectin-coated chitosan-LDH bionanocomposite beads for colon-targeted drug delivery", *Int. J. Pharm.* 463, 1– 9 (2014), to more recent ones by the same group (e.g., Rebitski et al., "Chitosan and pectin core-shell beads encapsulating metformin-clay intercalation compounds for controlled delivery", *New J. Chem.* 44, 10102–10110 (2020) where the systems is constructed several coatings to have a better control. None of this works have been cited or considered in this study but it would be relevant authors check them to clearly state the novelty of their approach as the formation of core-shell of that compositions for improve and control the delivery of oral drugs is clearly not new.

2) Once place the topic, I see the core-shell structures here developed seem to present a much smaller size than the core-shell beads on the reports above signaled but here, my question is, why would it be necessary to create very small nanoparticles for controlled release of oral administrate drugs? Actually, authors indicate in the text the possibility that the drug reach some organs in the body (they do not say which ones) but they do not offer any insides about how they expect the nanoparticles reach such organs. Or their system is intended for other type of administration?

- Secondly, I found other significant points for revision in the study that concerns to the following points:

1) What is the type of chitosan (molecular weight, as they are various) was used? How were prepared the solutions of different pH? How was the protocol to determine the efficiency of drug loading and what it was not introduced in the experimental section (there is some reference to it later in the results and discussion section (??))?

2) When described the protocol for the preparation of the core-shell nanoparticles authors use terms "solution" and "suspension" but it is hard to follow because they are not equivalent and sometimes are used to refer to the same system. Moreover, it must be noted that it is difficult to solve chitosan and in the text there is not explanation of the protocol followed to do it. Usually, chitosan is solved in acid to stabilize it protonated but in alcohol is hard to solve it, how do authors proceed in the present case? In Table 1 is signaled a composition of chitosan/pectin in the various systems, how were they determined? I cannot see any description of protocol indicating how the composition was analyzed.

3) There are few details of the protocol related to the study of drug delivery. Moreover, there some small indication regarding the RCP1 system but none to the RCP2 and RCP3. The same applies to the protocol regarding the antioxidant activity, only conditions related to the RCP1 system have been signaled. In this part again authors describe they prepare a "methanolic solution of RCP1", how is it possible to speak of solve the nanoparticles?

4) The FTIR characterization is incorrectly used the term peak when actually the correct one is band (something similar was later use when addressing UV-vis spectra). Authors must take care and use appropriate nomenclature when described signal in spectroscopic studies. Additionally, it is not clear the assignation of bands to the resveratrol in the RCP1 system as, apparently, the major components in the core-shell nanoparticles are the biopolymers. So, the more intense bands in the 1500-500 cm^{-1} region probably correspond to bands of the polymers instead that can show shifts in their frequencies due to the existence of different interactions than in the pure biopolymers. Authors should check carefully this part to be sure of the interpretation they afford.

4. Images in Fig.2 are of very poor quality and it I not possible to see the scale so I am not able to distinguish anything. Anyway, the indication the particles are of 23-32 nm seems to be hard to

ascertain based on them. Additionally, the EDX seems to be point less when the three components in the system are organic compounds. What was it used for?

5. The FETEM study seems to be applied to determine and see the size, shape and structure of the nanoparticles but from the afforded images it seems than only the images in Figure 3a, corresponding to the RCP1 system look like core-shell nanoparticles, the images of the other systems show the presence of particles with not well define shapes and structures. By the way, images in Fig. 3b and 3c seem to be duplicate. From those images it is not possible to see how the values of shell thickness and average diameter of the particles in Table 1 were established. Moreover, how it is possible to determine values with a precision of 0.01 nm from those images. From data in the Figure 4 it seems the particles are much larger than those authors said are the ones deduced from the TEM, at this point it is hard to believe that the agglomerates that suggest figures in DLS graphics were constituted by hundreds of nanoparticles (differences from 20 nm to > 1000 nm).

6. Interpretation of zeta potential is not clear, why authors say the values in system with ratio 1:15 indicated good stability, in relation to what? From this analysis is not clear if authors understand what is the information this technique affords and so their interpretation is totally unclear.

7. There is a section named as "Particle yield and Reveratrol loading efficiency" which describe some experiments that should be better included as an experimental section although it actually does not introduce a complete protocol but just certain information of how the loading of resveratrol was estimated. The information is not complete and also introduces doubts about the efficiency of the method as it is indicated the system was centrifuged at 1000 rpm for 10 minutes and it is hard to believe that in this situation was possible to separate nanoparticles as smaller as 20 nm as authors claim they have. Other points in this study is what there are only results for the RCP1 system and not to the other ones.

8. I will not introduce any comment on the drug delivery system study as far as I have not clear idea of what is the actual goal of the prepared systems, oral administration? "targets different organ sites" as signaled in the text? If it is for oral administration it is not possible to understand the experiments of release at different pH was not done in a sequential order as the one of the changes in the gastro-intestinal tract or why the release is kept till 30 hours when the maximum is much shorter. If the intended application is to introduce the nanoparticles in the organs then the essays have to be explained and designed more clearly.

9. If the main conclusion of the works is just a "pectin coating can serve as an effective strategy to stabilize chitosan loaded nanoparticles as oral delivery vehicles" as main outcome of the study, then it must be signaled that this is not something new. In fact, previous results by other authors, as I had signaled above already reported on it, and therefore, other points should be strengthen to justify the study.

10. Finally, it must be signaled the manuscript introduces diverse errata and typos that should be revised. Please, do not use capital letter when refers to common words such chitosan, pectin, etc.

Review form: Reviewer 5

Is the manuscript scientifically sound in its present form?

Yes

Are the interpretations and conclusions justified by the results?

Yes

Is the language acceptable?

Yes

Do you have any ethical concerns with this paper?

No

Have you any concerns about statistical analyses in this paper?

No

Recommendation?

Accept as is

Comments to the Author(s)

The manuscript is very interesting concerning the resveratrol loaded chitosan-pectin core-shell nanoparticles as novel drug delivery vehicle for sustained release and improved antioxidant activities. Most of the questions have been addressed. The reviewer suggest the acceptance of this manuscript. Advices to the authors, but not strongly recommended, especially about the DLS characterization:

As far as self-assembly of polymeric molecules in different solvents is commonly studied by DLS and the aggregation process can be investigated using this method, Although the particle sizes of TEM must be smaller than that of DLS, but the intensity sizes were much too different from the results of TEM. As usual, the more uniform of the distribution of nanoparticles, the closer of the number, intensity and volume sizes were. Therefore I suggest the authors provide the volume (%) sizes of nanoparticle, which are more persuasive than the intensity size.

Decision letter (RSOS-210784.R1)

Dear Dr Ganguly:

Title: Resveratrol Loaded Chitosan-Pectin Core-Shell Nanoparticles as Novel Drug Delivery Vehicle for Sustained Release and Improved Antioxidant Activities
Manuscript ID: RSOS-210784.R1

Thank you for submitting the above manuscript to Royal Society Open Science. On behalf of the Editors and the Royal Society of Chemistry, I am pleased to inform you that your manuscript will be accepted for publication in Royal Society Open Science subject to minor revision in accordance with the referee suggestions. Please find the reviewers' comments at the end of this email.

The reviewers and handling editors have recommended publication, but also suggest some minor revisions to your manuscript. Therefore, I invite you to respond to the comments and revise your manuscript.

Please also include the following statements alongside the other end statements. As we cannot publish your manuscript without these end statements included, if you feel that a given heading is not relevant to your paper, please nevertheless include the heading and explicitly state that it is

not relevant to your work. We have included a screenshot example of the end statements for reference.

- Ethics statement

Please clarify whether you received ethical approval from a local ethics committee to carry out your study. If so please include details of this, including the name of the committee that gave consent in a Research Ethics section after your main text. Please also clarify whether you received informed consent for the participants to participate in the study and state this in your Research Ethics section.

OR

Please clarify whether you obtained the necessary licences and approvals from your institutional animal ethics committee before conducting your research. Please provide details of these licences and approvals in an Animal Ethics section after your main text.

OR

Please clarify whether you obtained the appropriate permissions and licences to conduct the fieldwork detailed in your study. Please provide details of these in your methods section.

- Data accessibility

It is a condition of publication that you make available the data and research materials supporting the results in the article. Datasets should be deposited in an appropriate publicly available repository and details of the associated accession number, link or DOI to the datasets must be included in the Data Accessibility section of the article

(<https://royalsocietypublishing.org/rsos/for-authors#question17>). Reference(s) to datasets should also be included in the reference list of the article with DOIs (where available).

Please include a Data Availability section after your main text stating where supporting data are available from, or where they will be made available should your article be accepted for publication.

If you wish to submit your supporting data or code to Dryad (<http://datadryad.org/>), or modify your current submission to dryad, please use the following link:
<http://datadryad.org/submit?journalID=RSOS&manu=RSOS-210784.R1>

- Competing interests

Please include a Competing Interests section after your main text declaring any financial or non-financial competing interests. If you have no competing interests please state 'I/we have no competing interests.'

- Authors' contributions

Please include an Authors' Contributions section at the end of your main text detailing the contribution of each author. All authors should have read and approved the manuscript before submission and this should be stated in the Authors' Contributions section.

The list of Authors should meet all of the following criteria; 1) substantial contributions to conception and design, or acquisition of data, or analysis and interpretation of data; 2) drafting the article or revising it critically for important intellectual content; and 3) final approval of the version to be published.

- Acknowledgements

- Funding statement

Please include a funding section after your main text which lists the source of funding for each author.

Because the schedule for publication is very tight, it is a condition of publication that you submit the revised version of your manuscript before 13-Nov-2021. Please note that the revision deadline will expire at 00.00am on this date. If you do not think you will be able to meet this date please let me know immediately.

Supplementary files will be published alongside the paper on the journal website and posted on the online figshare repository (<https://figshare.com>). The heading and legend provided for each supplementary file during the submission process will be used to create the figshare page, so please ensure these are accurate and informative so that your files can be found in searches. Files

on figshare will be made available approximately one week before the accompanying article so that the supplementary material can be attributed a unique DOI.

Kind regards,
Dr Ellis Wilde
Publishing Editor, Journals

On behalf of the Subject Editor Professor Anthony Stace and the Associate Editor Professor Chaohua Cui.

RSC Associate Editor
Comments to the Author:
(There are no comments.)

RSC Subject Editor
Comments to the Author:
(There are no comments.)

Reviewer comments to Author:
Reviewer: 3
Comments to the Author(s)
All comments were viewed and responded to by authors but the quality of figures is not good and they must be improved.

Reviewer: 4
Comments to the Author(s)
This manuscript introduces the development of core-shell nanoparticles to be used in the controlled release of resveratrol. The topic could be interesting however, there are significant drawbacks in the reported study that precludes its recommendation for being published. I will signal some of the most significant ones with the aim help authors to improve their work before intend its publication:
The study seems to address the preparation of nanoparticles formed by a core of chitosan and an external coating of pectin to be used as a controlled drug delivery for oral administration.

- Firstly, I have various issues that arise me doubts about the goal, adequate situation of the study in relation to previous ones and the validation of the study with respect the final goal:

1) There are reports in the literature addressing the fabrication of core-shell structures of similar structure for oral drug administration, from the report by Ribeiro et al. "Pectin-coated chitosan-LDH bionanocomposite beads for colon-targeted drug delivery", *Int. J. Pharm.* 463, 1- 9 (2014), to more recent ones by the same group (e.g., Rebitski et al., "Chitosan and pectin core-shell beads encapsulating metformin-clay intercalation compounds for controlled delivery", *New J. Chem.* 44, 10102-10110 (2020) where the systems is constructed several coatings to have a better control. None of this works have been cited or considered in this study but it would be relevant authors check them to clearly state the novelty of their approach as the formation of core-shell of that compositions for improve and control the delivery of oral drugs is clearly not new.

2) Once place the topic, I see the core-shell structures here developed seem to present a much smaller size than the core-shell beads on the reports above signaled but here, my question is, why would it be necessary to create very small nanoparticles for controlled release of oral administrate drugs? Actually, authors indicate in the text the possibility that the drug reach some organs in the body (they do not say which ones) but they do not offer any insides about how they expect the nanoparticles reach such organs. Or their system is intended for other type of administration?

- Secondly, I found other significant points for revision in the study that concerns to the following points:

1) What is the type of chitosan (molecular weight, as they are various) was used? How were prepared the solutions of different pH? How was the protocol to determine the efficiency of drug loading and what it was not introduced in the experimental section (there is some reference to it later in the results and discussion section (??))?

2) When described the protocol for the preparation of the core-shell nanoparticles authors use terms "solution" and "suspension" but it is hard to follow because they are not equivalent and sometimes are used to refer to the same system. Moreover, it must be noted that it is difficult to solve chitosan and in the text there is not explanation of the protocol followed to do it. Usually, chitosan is solved in acid to stabilize it protonated but in alcohol is hard to solve it, how do authors proceed in the present case? In Table 1 is signaled a composition of chitosan/pectin in the various systems, how were they determined? I cannot see any description of protocol indicating how the composition was analyzed.

3) There are few details of the protocol related to the study of drug delivery. Moreover, there some small indication regarding the RCP1 system but none to the RCP2 and RCP3. The same applies to the protocol regarding the antioxidant activity, only conditions related to the RCP1 system have been signaled. In this part again authors describe they prepare a "methanolic solution of RCP1", how is it possible to speak of solve the nanoparticles?

4) The FTIR characterization is incorrectly used the term peak when actually the correct one is band (something similar was later use when addressing UV-vis spectra). Authors must take care and use appropriate nomenclature when described signal in spectroscopic studies. Additionally, it is not clear the assignation of bands to the resveratrol in the RCP1 system as, apparently, the major components in the core-shell nanoparticles are the biopolymers. So, the more intense bands in the 1500-500 cm^{-1} region probably correspond to bands of the polymers instead that can show shifts in their frequencies due to the existence of different interactions than in the pure biopolymers. Authors should check carefully this part to be sure of the interpretation they afford.

4. Images in Fig.2 are of very poor quality and it I not possible to see the scale so I am not able to distinguish anything. Anyway, the indication the particles are of 23-32 nm seems to be hard to ascertain based on them. Additionally, the EDX seems to be point less when the three components in the system are organic compounds. What was it used for?

5. The FETEM study seems to be applied to determine and see the size, shape and structure of the nanoparticles but from the afforded images it seems that only the images in Figure 3a, corresponding to the RCP1 system look like core-shell nanoparticles, the images of the other systems show the presence of particles with not well defined shapes and structures. By the way, images in Fig. 3b and 3c seem to be duplicate. From those images it is not possible to see how the values of shell thickness and average diameter of the particles in Table 1 were established. Moreover, how it is possible to determine values with a precision of 0.01 nm from those images. From data in the Figure 4 it seems the particles are much larger than those authors said are the ones deduced from the TEM, at this point it is hard to believe that the agglomerates that suggest figures in DLS graphics were constituted by hundreds of nanoparticles (differences from 20 nm to > 1000 nm).

6. Interpretation of zeta potential is not clear, why authors say the values in system with ratio 1:15 indicated good stability, in relation to what? From this analysis is not clear if authors understand what is the information this technique affords and so their interpretation is totally unclear.

7. There is a section named as "Particle yield and Resveratrol loading efficiency" which describes some experiments that should be better included as an experimental section although it actually does not introduce a complete protocol but just certain information of how the loading of resveratrol was estimated. The information is not complete and also introduces doubts about the efficiency of the method as it is indicated the system was centrifuged at 1000 rpm for 10 minutes and it is hard to believe that in this situation was possible to separate nanoparticles as small as 20 nm as authors claim they have. Other points in this study is what there are only results for the RCP1 system and not to the other ones.

8. I will not introduce any comment on the drug delivery system study as far as I have not clear idea of what is the actual goal of the prepared systems, oral administration? "targets different organ sites" as signaled in the text? If it is for oral administration it is not possible to understand the experiments of release at different pH was not done in a sequential order as the one of the changes in the gastro-intestinal tract or why the release is kept till 30 hours when the maximum is much shorter. If the intended application is to introduce the nanoparticles in the organs then the essays have to be explained and designed more clearly.

9. If the main conclusion of the work is just a "pectin coating can serve as an effective strategy to stabilize chitosan loaded nanoparticles as oral delivery vehicles" as main outcome of the study, then it must be signaled that this is not something new. In fact, previous results by other authors, as I had signaled above already reported on it, and therefore, other points should be strengthened to justify the study.

10. Finally, it must be signaled the manuscript introduces diverse errata and typos that should be revised. Please, do not use capital letter when refers to common words such chitosan, pectin, etc.

Reviewer: 5

Comments to the Author(s)

The manuscript is very interesting concerning the resveratrol loaded chitosan-pectin core-shell nanoparticles as novel drug delivery vehicle for sustained release and improved antioxidant activities. Most of the questions have been addressed. The reviewer suggests the acceptance of this manuscript. Advices to the authors, but not strongly recommended, especially about the DLS characterization:

As far as self-assembly of polymeric molecules in different solvents is commonly studied by DLS and the aggregation process can be investigated using this method, Although the particle sizes of

TEM must be smaller than that of DLS, but the intensity sizes were much too different from the results of TEM. As usual, the more uniform of the distribution of nanoparticles, the closer of the number, intensity and volume sizes were. Therefore I suggest the authors provide the volume (%) sizes of nanoparticle, which are more persuasive than the intensity size.

Author's Response to Decision Letter for (RSOS-210784.R1)

See Appendix C.

RSOS-210784.R2

Review form: Reviewer 4

Is the manuscript scientifically sound in its present form?

No

Are the interpretations and conclusions justified by the results?

No

Is the language acceptable?

Yes

Do you have any ethical concerns with this paper?

No

Have you any concerns about statistical analyses in this paper?

No

Recommendation?

Major revision is needed (please make suggestions in comments)

Comments to the Author(s)

I have read with attention the response to my queries and the revised manuscript and I regret to express that some of the most relevant points I signaled in my previous report were not at all understood in some cases, and in other just skipped. Now at least I can understand better the goal of the study in relation to the target of delivery so, it makes clearer that some of my previous questions were not addressed in the revised manuscript that, by the way, introduces just a few changes. As I had pointed out in my previous report, the development of chitosan-pectin core-shell nanoparticles to produce a drug delivery systems for oral administration of resveratrol is new, but the use of chitosan-pectin core-shell structures to produce drug delivery systems for oral administration of drugs is not new. Amongst the diverse reports in the I had signaled two examples (one selected because it was the first publication on the topic and the other one as a specific example in which the system is improved to oral delivery a drug that present problems for being adsorbed, as in the present case), for introducing the topic and to place the novelty of the here reported approach in context. Therefore, this should be stressed in the introduction not somewhere else within the results and discussion section because the approach here used for oral drug delivery is not at all new even though it is now applied to delivery another specific drug,

resveratrol, and the produced spheres are nanometric instead of micro or milimetric as in other approaches. The second point that arisen some concerns to me is related to the precise mechanism of action of the produced drug delivery system. From the answer authors made to my first query, it looks that the produced nanospheres will be adsorbed at the intestine as they are and then will be “move to liver and then to the specific organ through systemic circulation” (sic). This point is relevant and the manuscript doesn't offer proof it could work from a physiological point of view, is there any evidence with similar type of systems? In general, drug delivery systems for actions at specific location of tumors use to be delivery via intravenous to assure it can reach the target, some references to support they

Regarding my other queries, some of them have been explained and revised except other ones that I consider relevant and, mainly they have been just ignored:

- I insist, the EDX study is pointless when the three components in the system are organic compounds. What were the expected “other elements” to be concerned in relation to the “purity of the samples”?

- I do not see any response in relation to the proposed protocol of separation of nanoparticles using just centrifugation at 1000 rpm for 10 minutes.

- The study of drug delivery at 3 pH but without a sequential protocol similar to the gastro-intestinal tract is just informative but not conclusive when the intended application is a drug for oral administration. The results could inform of the stability and possible release in the different conditions but, to be more realistic, it is necessary to see what happens when the same system evolves from one to other pH medium. Additionally, I am quite surprised that, with the small size of the nanoparticles used in the study, after 30 hours the release was not complete, as the biopolymers may dissolve, as least partially. Is there any evidence of the same of the nanoparticles after the release study?

- Finally, I have still concerns in relation to the statement that “the prepare nanoparticles can better penetrate the tumor cells, etc.” authors introduced for answering my query related to the conclusions part. From the introduced results, the system could be promised for it but the manuscript does not show conclusive evidence of it. Therefore, I would be more careful at this respect.

Decision letter (RSOS-210784.R2)

Dear Dr Ganguly:

Title: Resveratrol Loaded Chitosan-Pectin Core-Shell Nanoparticles as Novel Drug Delivery Vehicle for Sustained Release and Improved Antioxidant Activities
Manuscript ID: RSOS-210784.R2

Thank you for submitting the above manuscript to Royal Society Open Science. On behalf of the Editors and the Royal Society of Chemistry, I am pleased to inform you that your manuscript will be accepted for publication in Royal Society Open Science subject to minor revision in accordance with the referee suggestions. Please find the reviewers' comments at the end of this email.

The reviewers and handling editors have recommended publication, but also suggest some minor revisions to your manuscript. Therefore, I invite you to respond to the comments and revise your manuscript.

Please also include the following statements alongside the other end statements. As we cannot publish your manuscript without these end statements included, if you feel that a given heading is not relevant to your paper, please nevertheless include the heading and explicitly state that it is not relevant to your work. We have included a screenshot example of the end statements for reference.

- Ethics statement

Please clarify whether you received ethical approval from a local ethics committee to carry out your study. If so please include details of this, including the name of the committee that gave consent in a Research Ethics section after your main text. Please also clarify whether you received informed consent for the participants to participate in the study and state this in your Research Ethics section.

OR

Please clarify whether you obtained the necessary licences and approvals from your institutional animal ethics committee before conducting your research. Please provide details of these licences and approvals in an Animal Ethics section after your main text.

OR

Please clarify whether you obtained the appropriate permissions and licences to conduct the fieldwork detailed in your study. Please provide details of these in your methods section.

- Data accessibility

It is a condition of publication that you make available the data and research materials supporting the results in the article. Datasets should be deposited in an appropriate publicly available repository and details of the associated accession number, link or DOI to the datasets must be included in the Data Accessibility section of the article (<https://royalsocietypublishing.org/rsos/for-authors#question17>). Reference(s) to datasets should also be included in the reference list of the article with DOIs (where available).

Please include a Data Availability section after your main text stating where supporting data are available from, or where they will be made available should your article be accepted for publication.

If you wish to submit your supporting data or code to Dryad (<http://datadryad.org/>), or modify your current submission to dryad, please use the following link:
<http://datadryad.org/submit?journalID=RSOS&manu=RSOS-210784.R2>

- Competing interests

Please include a Competing Interests section after your main text declaring any financial or non-financial competing interests. If you have no competing interests please state 'I/we have no competing interests.'

- Authors' contributions

Please include an Authors' Contributions section at the end of your main text detailing the contribution of each author. All authors should have read and approved the manuscript before submission and this should be stated in the Authors' Contributions section.

The list of Authors should meet all of the following criteria; 1) substantial contributions to conception and design, or acquisition of data, or analysis and interpretation of data; 2) drafting the article or revising it critically for important intellectual content; and 3) final approval of the version to be published.

- Acknowledgements

- Funding statement

Please include a funding section after your main text which lists the source of funding for each author.

Because the schedule for publication is very tight, it is a condition of publication that you submit the revised version of your manuscript before 12-Dec-2021. Please note that the revision deadline will expire at 00.00am on this date. If you do not think you will be able to meet this date please let me know immediately.

Supplementary files will be published alongside the paper on the journal website and posted on the online figshare repository (<https://figshare.com>). The heading and legend provided for each supplementary file during the submission process will be used to create the figshare page, so

please ensure these are accurate and informative so that your files can be found in searches. Files on figshare will be made available approximately one week before the accompanying article so that the supplementary material can be attributed a unique DOI.

Kind regards,
Dr Ellis Wilde
Publishing Editor, Journals

On behalf of the Subject Editor Professor Anthony Stace and the Associate Editor Professor Chaohua Cui.

RSC Associate Editor

Comments to the Author:

The article published in Royal Society Open Science should be scientifically sound, in which the methodology is rigorous and the conclusions are fully supported by the data. In light of the comments, I therefore invite you to provide a point-by-point response letter addressed to the reviewer, including a list of changes made and a rebuttal to any comments with which you disagree.

RSC Subject Editor

Comments to the Author:

(There are no comments.)

Reviewer comments to Author:

Reviewer: 4

Comments to the Author(s)

I have read with attention the response to my queries and the revised manuscript and I regret to express that some of the most relevant points I signaled in my previous report were not at all understood in some cases, and in other just skipped. Now at least I can understand better the goal of the study in relation to the target of delivery so, it makes clearer that some of my previous questions were not addressed in the revised manuscript that, by the way, introduces just a few changes. As I had pointed out in my previous report, the development of chitosan-pectin core-shell nanoparticles to produce a drug delivery system for oral administration of resveratrol is new, but the use of chitosan-pectin core-shell structures to produce drug delivery systems for oral administration of drugs is not new. Amongst the diverse reports that I had signaled two examples (one selected because it was the first publication on the topic and the other one as a specific example in which the system is improved to oral delivery of a drug that presents problems

for being adsorbed, as in the present case), for introducing the topic and to place the novelty of the here reported approach in context. Therefore, this should be stressed in the introduction not somewhere else within the results and discussion section because the approach here used for oral drug delivery is not at all new even though it is now applied to delivery another specific drug, resveratrol, and the produced spheres are nanometric instead of micro or milimetric as in other approaches. The second point that arisen some concerns to me is related to the precise mechanism of action of the produced drug delivery system. From the answer authors made to my first query, it looks that the produced nanospheres will be adsorbed at the intestine as they are and then will be “move to liver and then to the specific organ through systemic circulation” (sic). This point is relevant and the manuscript doesn’t offer proof it could work from a physiological point of view, is there any evidence with similar type of systems? In general, drug delivery systems for actions at specific location of tumors use to be delivery via intravenous to assure it can reach the target, some references to support they

Regarding my other queries, some of them have been explained and revised except other ones that I consider relevant and, mainly they have been just ignored:

- I insist, the EDX study is pointless when the three components in the system are organic compounds. What were the expected “other elements” to be concerned in relation to the “purity of the samples”?

- I do not see any response in relation to the proposed protocol of separation of nanoparticles using just centrifugation at 1000 rpm for 10 minutes.

- The study of drug delivery at 3 pH but without a sequential protocol similar to the gastro-intestinal tract is just informative but not conclusive when the intended application is a drug for oral administration. The results could inform of the stability and possible release in the different conditions but, to be more realistic, it is necessary to see what happens when the same system evolves from one to other pH medium. Additionally, I am quite surprised that, with the small size of the nanoparticles used in the study, after 30 hours the release was not complete, as the biopolymers may dissolve, as least partially. Is there any evidence of the same of the nanoparticles after the release study?

- Finally, I have still concerns in relation to the statement that “the prepare nanoparticles can better penetrate the tumor cells, etc.” authors introduced for answering my query related to the conclusions part. From the introduced results, the system could be promised for it but the manuscript does not show conclusive evidence of it. Therefore, I would be more careful at this respect.

Author's Response to Decision Letter for (RSOS-210784.R2)

See Appendix D.

RSOS-210784.R3

Review form: Reviewer 4

Is the manuscript scientifically sound in its present form?

Yes

Are the interpretations and conclusions justified by the results?

Yes

Is the language acceptable?

Yes

Do you have any ethical concerns with this paper?

No

Have you any concerns about statistical analyses in this paper?

No

Recommendation?

Accept as is

Comments to the Author(s)

I have checked the new revised manuscript and the responses to my last queries. I see authors keep introducing answers addressed to me but, actually, very few changes in the manuscript in relation to the queries to clarify and support their points. At this stage, I consider the explanations are ok so I will not go further and if the Editor of the journal consider the manuscript is ready for being accepted is ok from my side.

Decision letter (RSOS-210784.R3)

Dear Dr Ganguly:

Title: Resveratrol Loaded Chitosan-Pectin Core-Shell Nanoparticles as Novel Drug Delivery Vehicle for Sustained Release and Improved Antioxidant Activities
Manuscript ID: RSOS-210784.R3

It is a pleasure to accept your manuscript in its current form for publication in Royal Society Open Science. The chemistry content of Royal Society Open Science is published in collaboration with the Royal Society of Chemistry.

Yours sincerely,
Dr Ellis Wilde
Publishing Editor, Journals

On behalf of the Subject Editor Professor Anthony Stace and the Associate Editor Professor Chaohua Cui.

RSC Associate Editor
Comments to the Author:
(There are no comments.)

RSC Subject Editor
Comments to the Author:
(There are no comments.)

Reviewer(s)' Comments to Author:
Reviewer: 4

Comments to the Author(s)

I have checked the new revised manuscript and the responses to my last queries. I see authors keep introducing answers addressed to me but, actually, very few changes in the manuscript in relation to the queries to clarify and support their points. At this stage, I consider the explanations are ok so I will not go further and if the Editor of the journal consider the manuscript is ready for being accepted is ok from my side.

Appendix A

Manuscript Id: RSOS-210784

Recommendations: Major revision

1. Authors should improve image quality of the scheme
2. Figure 1 should be incorporated into the results section
3. For control, FTIR spectra of the free chitosan, pectin and resveratrol should be provided and the suitable discussion must be added
4. Kindly provide the experimental details of FTIR sample preparation
5. Improve the image quality of TEM images so as to clearly highlight the scale bars
6. Perform other size measurement studies such as light scattering, and also zeta potential measurement. Stability of the nanoparticles in the suitable biological media should also be mentioned
7. Error bars and statistical significance in figure 5a should be incorporated and fig 6
8. Authors should discuss the role of different ratios of the polysaccharides in determining the kinetics of release
9. Overall discussion of the manuscript is below the currently accepted level of other publications. Authors should work in this aspect and should include some more discussions
10. How did authors determine the extent of pectin coating over the chitosan nanoparticles?
11. If possible, authors should also include preliminary cell studies so as to increase the overall impact of the work

Appendix B

Response to the referee's comments

Journal Name: - RSOS (Royal Society Open Science)

Manuscript ID: RSOS-210784

TITLE: Resveratrol Loaded Chitosan-Pectin Core-Shell Nanoparticles as Novel Drug Delivery Vehicle for Sustained Release and Improved Antioxidant Activities

Dear Editor,

Thank you for suggesting necessary revisions in our manuscript. The manuscript has been modified based on the suggestions and comments made by the reviewers. Detailed corrections and changes in manuscript are listed below point by point:

Reviewers' Comments to Author:

Reviewer: 1

Comments to the Author(s)

1. The *in vitro* drug release from the prepared core-shell nanoparticles was monitored with the help of a UV-Visible spectrophotometer in the absorbance was checked at 285 nm. Does this mean that the resveratrol used is *cis*-resveratrol? The material and method only convey the purity and origin of the resveratrol used. While in schematic A you write down the structure of the compound of *trans*-resveratrol. The value of the maximum lambda of *trans*-resveratrol, not 285 nm.

Our response:

The resveratrol used in the experiments was the *trans* isomer. But on exposure to sunlight, irradiation with UV rays, and when aqueous buffer medium is used to measure the absorption, the *trans* isomer of resveratrol is converted into *cis* isomer (as supported by reference no. ...) and a shift of the absorption maxima towards shorter wavelength is observed. A clearly visible shift in the wavelength absorbed by *trans* resveratrol from 304 to 285 nm on exposure to light was also reported by Camont *et. al* (Ref). This is why the absorbance was monitored at 285 nm and not at around 304 nm (which

is characteristic peak shown by trans resveratrol in ethanol or DMSO) for the in vitro drug release from the prepared core-shell nanoparticles.

In materials and methods, the specific name of the isomer of resveratrol (3, 5, 4'-trihydroxy-trans-stilbene) has been inserted.

Following text with figure has been inserted in the manuscript in results and discussion part in response to the above query-

Trans isomer of resveratrol was used in the samples of core – shell nanoparticles prepared for drug delivery. But on UV-irradiation, the trans isomer is converted into cis isomer (29, 30). Figure 5 shows the absorbance peak at 285 nm due to cis resveratrol released by RCP1. However, the peak due to trans resveratrol was obtained as only a low intense shoulder in between 300-350nm. This is why the absorbance was checked at 285 nm and not at around 304 nm for the in vitro drug release from the prepared core-shell nanoparticles.

Figure 5 UV- Vis spectra of resveratrol released from RCP1

Reference:

29. Kerrilee E. Allan, Claire E. Lenehan and Amanda V. Ellis, 2009. UV Light Stability of α -Cyclodextrin/Resveratrol Host–Guest Complexes and Isomer Stability at Varying pH, *Aust. J. Chem*, 62, 921–926. (doi:10.1071/CH08506)

30. Laurent C, Charles-Henry C, Yara R, Valérie NA, Raja D, Fabrice C, Jean-Louis B; Dominique BR, 2009. Simple spectrophotometric assessment of the trans-/cis-

resveratrol ratio in aqueous solutions. *Anal. Chim. Acta.* 634(1), 121–128. (doi: 10.1016/j.aca.2008.12.003)

2. The length of the introduction should be reduced. It is too lengthy. It would be better if only the important points were written down and some were transferred to the results and discussion.

Our response:

The introduction part has been shortened from 734 words to 529 words and has been rewritten in order to include the important points only.

The revised introduction part as inserted in the manuscript is as follows-

One of the prime attributes of an efficient drug-delivery vehicle is the controlled drug release rate. To increase the efficiency and therapeutic activity, the drug-delivery system should be made trigger dependent with sustained release behaviour. For this, various multifunctional nanoparticles with multiple triggers and complex structures have been developed (1, 2). Nowadays, nano carriers are being frequently used for drug delivery. These nanocarriers can decrease the frequency of administration while maintaining the steady and effective concentration of the drug at the target site (3, 4). Several nanoparticle-based products for diagnostics and therapeutics have been clinically approved and many of them are currently under clinical trials (5). During the last few decades, the main focus was on the development of biodegradable nanoparticles for effective drug delivery (6). Among various classes of nanoparticles, the core-shell is the most promising class with various biomedical applications. On account of surface chemistry its affinity to bind with drugs, receptors, ligands, etc. is very high. (7). Natural biopolymers such as pectin, alginate and chitosan have recently found wide applications in food and pharmaceutical sectors (8). The presence of functional groups in the biopolymer structures not only helps to encapsulate bioactive compounds or drugs but also to form a cross-linked dense network after encapsulation (9).

Recently pectin-based delivery systems have been registered for colon-specific delivery purposes. Pectin is a natural, anionic polysaccharide obtained from the cell walls of most fruits viz, apples, oranges and pears (10). Though it is resistant to the enzymes present in upper gastro-intestinal tract, it undergoes complete degradation by colonic bacterial enzymes (11). Chitosan is another linear polysaccharide with cationic properties which is obtained from chitin, a major component of the shells of shrimp, lobsters and crabs. Pectin and chitosan are known to combine to form a polyelectrolyte complex which can be used to encapsulate bioactive molecules and drugs. (12)

Resveratrol is a polyphenolic compound found in red grapes, cranberries and blueberries that possesses a wide range of beneficial effects on human health (13,14). It is found in two different structural configuration, cis-Resveratrol and trans-Resveratrol. Trans-configuration is the dominant configuration which represents the most thoroughly studied chemical form (15). Recently, considerable attention has been paid to the molecule as several health benefits of resveratrol, such as anti-

inflammatory activity, hepatoprotection, cardio and neuro-protection, anti-cancer activity, anti-aging effects, and diabetes prevention (16, 17) are reported. Despite the multiple health benefits of this natural molecule, it is often difficult to incorporate resveratrol into commercial pharmaceutical products because its poor water solubility and chemical instability lowers the bioavailability (18, 19). Moreover, free resveratrol gets metabolized quite rapidly in biological systems.

The benefits of resveratrol are only realizable when it is delivered in an encapsulated form in the body. However, this approach has not been fully explored and only few studies have been reported in this area.

Therefore, the aim of the present study is to develop chitosan-pectin core shell nanoparticles loaded with resveratrol in order to enhance its bioavailability and then analyse the effect of thickness of the pectin shell on the drug release efficiency of the formulation. It is also intended to investigate the effect of encapsulation on the radical scavenging activity of resveratrol.

Reviewer: 2

Comments to the Author(s)

1. Authors should improve image quality of the scheme

Our response:

The image quality of the scheme as well as graphical abstract has been improved and inserted in the revised manuscript as advised. The improved scheme is as follows-

Schematic A: Fabrication of Resveratrol loaded Chitosan-Pectin Core-Shell Nanoparticles

The graphical abstract is as follows

2. Figure 1 should be incorporated into the results section

Our Response:

Figure 1 has been incorporated in results section as figure 5.

3. For control, FT-IR spectra of the free chitosan, pectin and resveratrol should be provided and the suitable discussion must be added.

Our Response:

The FT-IR spectra of free chitosan, pectin and resveratrol have been provided and discussed as advised.

Following figures and text have been inserted in the revised manuscript –

The FT-IR spectra of free chitosan, pectin and resveratrol have been recorded for control. The FT-IR spectra of chitosan (figure 1a) showed a broad band at 3451 cm^{-1} for N-H symmetrical stretch overlapped with O-H stretching band. The peak at 2925 cm^{-1} was due to C-H stretching. The Chitosan

polymer shows characteristic band of amide I (C=O stretch) at 1637cm^{-1} and amide II (C-N stretching and N-H bending) at 1542cm^{-1} . The peak for amide III (N-H deformation vibration coupled with C-N stretching) is observed at 1374cm^{-1} (21). Another peak at 1160cm^{-1} is due to C-O-C stretching of the saccharide units. The peak at 1050cm^{-1} is for NH bending vibration.

The FT-IR Spectra of Pectin (figure 1 b) shows a broad peak at 3410cm^{-1} due to O-H stretching. The peak at 2925cm^{-1} is due to C-H stretching. The polymer shows a typical peak at 1754cm^{-1} due to the presence of ester carbonyl C=O group (22). Peaks at 1644cm^{-1} and 1408cm^{-1} correspond to the COO-group stretching (range $1600\text{-}1650\text{cm}^{-1}$ for antisymmetric stretch and $1400\text{-}1450\text{cm}^{-1}$ for symmetric stretch). Another peak at 1102cm^{-1} is observed due to the presence of ether linkage and glycosidic bond in pectin. An absorption peak at 1060cm^{-1} is observed due to C-H bending.

The FT-IR Spectra of Resveratrol (figure 1 c) showed a narrow band at 3298cm^{-1} for phenolic O-H stretching. There are several intense bands in 1650cm^{-1} to 1000cm^{-1} region out of which the band at 1383cm^{-1} is due to C-O stretching of phenolic group. The band at 1586cm^{-1} correspond to C=C olefinic stretching and the band at 1606cm^{-1} is due to C=C aromatic double-bond stretching. The peak at 965cm^{-1} is the characteristic of trans olefinic stretch which confirms the presence of resveratrol in trans isomer form. The band at 831cm^{-1} is the characteristic of olefinic group (23, 24).

Figure 1: FT-IR spectra of (a) free chitosan (b) Pectin (c) Resveratrol (d) RCP1

The FTIR spectra of RCP1 (figure 1d) exhibited three strong absorption bands at 1637cm^{-1} , 1094cm^{-1} and 1340cm^{-1} due to C=C aromatic double bond stretching, C=C olefinic stretching, and C-C stretching respectively in Resveratrol (25, 26). The peak at 3450cm^{-1} has been attributed to the vibrational stretching of O-H bond of resveratrol loaded chitosan pectin core-shell nanoparticles. Peak at 685cm^{-1} corresponds to a C-H stretch from the aromatic ring. The comparison of FT-IR spectrum of

RCP1 with that of free Chitosan, pectin and resveratrol reveals that there is no major functional change in the components after encapsulation. However, the band at 1542 cm^{-1} in chitosan for amide II stretch was very much reduced in intensity along with the band at 1644 cm^{-1} for carboxyl group stretching in pectin. The intensity of the band at 1754 cm^{-1} due to methyl ester of pectin is also reduced to large extent indicating the intermolecular electrostatic interaction of carboxyl group of pectin with the amino group of chitosan.

The FT-IR spectra of RCP2 and RCP3 were also compared with the FT-IR spectra of RCP1 (figure ES1). No difference was found in the three spectra which indicates that the interaction between the polymers is not changed on varying the ratio of Chitosan and pectin in the prepared core shell nanoparticles.

4. Kindly provide the experimental details of FTIR sample preparation

Our response:

Following experimental details have been provided in Materials and Methods part of the revised manuscript-

All the samples were characterised by FT-IR spectroscopic method. For sample preparation, KBr pellet method was used. The sample was mixed with anhydrous potassium bromide in a ratio of 1:100 and grounded to fine powder in a mortar-pestle. The mixture was transferred into pellet press dies and pressed in a manual hydraulic pellet press machine to obtain a pellet of the sample mixed with KBr. The pellet was then placed in ALPHA Bruker FT-IR spectrometer in order to record the spectrum over a wave number range of 4000 cm^{-1} – 500 cm^{-1} .

5. Improve the image quality of TEM images so as to clearly highlight the scale bars

Our response:

The image quality of TEM images has been improved. The improved images have been inserted in the revised manuscript as follows -

Figure 3a: HRTEM/SAED images of RCP1

Figure3 b: FETEM/SAED images of RCP2

Figure 3c: HRTEM/SAED images of RCP3

6. Perform other size measurement studies such as light scattering, and also zeta potential measurement. Stability of the nanoparticles in the suitable biological media should also be mentioned

Our response:

The Dynamic Light Scattering (DLS) experiments and Zeta Potential measurements have been performed for 0.025% concentrations of each RCP1 and RCP2 and RCP3. Stability of the nanoparticles, as revealed from the zeta potential values, has also been mentioned in the discussion as follows-

Following text has been included in the materials and methods part-

Following text has been included in Results and Discussion part-

The hydrodynamic diameters of the resveratrol loaded chitosan-pectin core-shell nanoparticles were measured using Zetasizer Nano ZS90 instrument (model no. ZEN3690). For the sample preparation in DLS experiment, 0.025 g each of RCP1, RCP2 and RCP3 was added to deionized water and sonicated for 2 hours. The dispersion was then used for DLS measurements.

DLS (Dynamic Light Scattering) Experiment

The DLS experiments show the size distribution of the particles of RCP1, RCP2 and RCP3 with respect to intensity (figure 4a, 4b and 4c). The sizes (hydrodynamic diameter values) calculated from DLS studies were much higher than the sizes reported from TEM studies. This is not unusual because the TEM image gives the actual size of the particles determined on a statistically small sample whereas DLS provides the hydrodynamic diameter which is the diameter of a hypothetical sphere having similar diffusion coefficient in the same environment of the particles. When the nanoparticle is dispersed, its surface shows strong interactions with solvent molecules which strongly influences behaviour of the nanoparticles. The presence of surface charges on the particles causes repulsion between the particles and helps to prevent agglomeration.

Figure 4 (a): Hydrodynamic diameter of RCP1 measured using DLS

Figure 4 (b): Hydrodynamic diameter of RCP2 measured using DLS

Figure 4 (c): Hydrodynamic diameter of RCP3 measured using DLS

Zeta Potential Measurement

Zeta potential measurements of the three samples RCP1, RCP2 and RCP3 show the magnitude of the surface charge of the colloidal particles of each sample as given in (table 2)

Table 2: Zeta potential of the colloidal particles of RCP1, RCP2 and RCP3

Resveratrol loaded chitosan-pectin core-shell nanoparticles	RCP1	RCP2	RCP3
Chitosan: pectin	1:7	1:15	1:4
Zeta potential at a concentration of 0.025%	-12.2 mV	-47.7 mV	+ 10.8 mV

The magnitude of the surface charge was found to be higher in case of RCP2 which may be due to the thicker pectin shell with polar end groups. In case of RCP1 and RCP3, due to interaction between the two biopolymers (pectin and Chitosan), the charges are somewhat diminished depending upon the ratio of the two. This is why the zeta potential values are found to be low. The zeta potential value of the RCP2 having Chitosan: pectin in the ratio 1:15 indicated good stability (27) while the values of RCP1 and RCP3 indicate lower stability when compared to the other two. However, none of the values are suggestive of rapid agglomeration.

Though the dilute samples used for DLS and Zeta Potential measurements cannot adequately represent the therapeutic formulations to be used in vivo, they primarily indicate the potential of the use of the formula to design a drug delivery vehicle.

7. Error bars and statistical significance in figure 5a should be incorporated and fig 6

Our response:

The data for repeated drug delivery experiments for RCP1 at different pH values (figure 5a) and antioxidant activity experiments (figure 6a) have been used in order to insert error bars. The average and standard deviation for these data were calculated in Microsoft excel sheet and the graph were plotted in Microcal Origin 6.0 Professional software setting standard deviation data as Y axis error bar. The graphs which are figure 6(a) and figure 7 respectively in the revised manuscript are as follows-

Figure 6a: Time versus % Release graph of resveratrol from RCP1at various pH

Figure 7: DPPH radical scavenging percentage of ascorbic acid, free RSV and encapsulated RSV, RCP1 (in methanol) at different concentrations

Statistical analysis using two-way ANOVA indicates from the comparison of F values with F critical values and P values with α values that encapsulated resveratrol is significantly different and better than free RSV in antioxidant activity (Table ES1).

The above statistical significance of the data has been inserted in Electronic Supplementary Information.

8. Authors should discuss the role of different ratios of the polysaccharides in determining the kinetics of release

Our response:

Following text has been included in the discussion part-

The drug release efficiency was checked for both thick (RCP2) and thin shelled (RCP3) core shell nanoparticles and was compared with that of RCP1. Both RCP2 and RCP3 showed faster drug release in acidic and alkaline pH compared to the neutral medium as was observed with RCP1. However, the three samples were different in the percentage of drug delivered with time. In case of RCP1, less than 60% of the drug was released in the first 24 hours and the sustained release behaviour continued even up to 30 hours indicating that the drug is released from the matrix slowly. Moreover, the release of the drug from RCP1 followed the similar kinetics at all the pH ranges studied. In case of RCP2 (figure 6b), about 45% of the drug was released within first one hour in acidic medium but the rate of release became slower subsequently. The release pattern was found to be similar at other pH values of the medium. About 60% of the drug was released within 24 hours and then the release almost stopped. RCP3, however, showed more than 80% release of the drug within first four hours in acidic medium and 90% of the drug was released in 30 hours (figure 6c). In alkaline and neutral medium,

about 50% of the drug was released in first four hours and continued up to 60% in 30 hours showing better release profile.

Therefore, among the three samples RCP1, RCP2 and RCP3 having Chitosan and Pectin ratio 1:7, 1:15 and 1:4 respectively, RCP1 with medium pectin shell thickness was the best drug carrier with sustained release efficiency. The ratio of the two polysaccharides plays a significant role in the stability of the prepared samples and the kinetics of the drug release. When the ratio of the two polysaccharides, chitosan and pectin, is changed, the rate of drug release decreases with increasing thickness of pectin shell. However, the sustained release was observed when the ratio is 1:7. Therefore, the thickness of pectin shell in these core shell nanoparticles must be optimised taking into consideration the sustained release of drug from the newly designed drug delivery vehicle.

9. Overall discussion of the manuscript is below the currently accepted level of other publications. Authors should work in this aspect and should include some more discussions

Our Response:

Discussion part has been extended with more discussion on the FT-IR spectra, newly added DLS experiments with zeta potential measurements and their implications.

Pectin derivatives carry a high charge density (either positive primary amine or negative carboxyl groups) and are able to penetrate deeply into tissue. Formulations with pectin derivatives show better ability to retain the incorporated drugs. The pectin coating not only significantly increased the capability of chitosan nanoparticles to encapsulate resveratrol but also influenced the rate of release. Therefore, pectin coating can serve as an effective strategy to stabilize chitosan loaded drug nanoparticles as oral delivery vehicles.

10. How did authors determine the extent of pectin coating over the chitosan nanoparticles?

Our Response:

Following text has been incorporated in Materials and Methods part of the revised manuscript

The extent of pectin coating over the surface of resveratrol loaded chitosan nanoparticles was determined from HRTEM images of each RCP1, RCP2 and RCP3. Several images of each sample with wide distribution of core shell nanoparticles were selected. The size of several core-shell nanoparticle in a particular image was measured using the scale bar provided in that image. From the contrast difference of chitosan and pectin, the shell thickness was measured. The average value of the thickness of pectin of 20 to 30 core – shell nanoparticles of each sample was considered as final value.

11. If possible, authors should also include preliminary cell studies so as to increase the overall impact of the work.

The authors do not possess the necessary expertise to carry out preliminary cell studies to include in the present manuscript. However, we appreciate the suggestion of the reviewer. The authors will try to incorporate such studies in future manuscripts.

Appendix C

Response to the referee's comments

Journal Name: - RSOS (Royal Society Open Science)

Manuscript ID: RSOS-210784 R1

TITLE: Resveratrol Loaded Chitosan-Pectin Core-Shell Nanoparticles as Novel Drug Delivery Vehicle for Sustained Release and Improved Antioxidant Activities

Dear Editor,

Thank you for suggesting minor revisions to the manuscript. The manuscript has been modified based on the suggestions and comments made by the reviewers. Detailed corrections and changes in manuscript are listed below point by point:

Reviewers' Comments to Author:

Reviewer: 3

Comments to the Author(s)

All comments were viewed and responded to by authors but the quality of figures is not good and they must improve them.

Our response:

The quality of figures has been improved as suggested.

Reviewer: 4

1) There are reports in the literature addressing the fabrication of core-shell structures of similar structure for oral drug administration, from the report by Ribeiro et al. "Pectin-coated chitosan-LDH bionanocomposite beads for colon-targeted drug delivery", *Int. J. Pharm.* 463, 1–9 (2014), to more recent ones by the same group (e.g., Rebitski et al., "Chitosan and pectin core-shell beads encapsulating metformin-clay intercalation compounds for controlled delivery", *New J. Chem.* 44, 10102–10110 (2020) where the systems is constructed several coatings to have a better control. None of this works have been cited or considered in this study but it would be relevant authors check them to clearly state the novelty of their approach as the formation of core-shell of that compositions for improve and control the

delivery of oral drugs is clearly not new.

Our response:

The above mentioned two publications have been checked and cited in the revised manuscript stating the novelty of our approach.

Our work is mainly on the delivery of a specific drug, resveratrol, which in spite of its strong antioxidant and cancer preventing activity, is not effective as an oral drug due to poor bioavailability and rapid metabolism. The focus of our work was to design an effective nano carrier for this particular drug and not to design a universal carrier. As nanoparticles can better penetrate the tumour cells, the prepared compositions can be more effective in preventing and curing cancer.

Following text and references have been inserted in the revised manuscript-

Ribeiro et al. have reported the preparation of biohybrid beads coated with pectin and studied colon specific drug delivery taking 5-aminosalicylic acid as model drug (34). Rebitski et al. have also prepared the core-shell beads with clay as host into chitosan/pectin core and was utilized in the delivery of metformin (35). Our work, however, focuses mainly on a specific drug, resveratrol which in spite of its strong antioxidant and cancer preventing activity, is not effective as an oral drug due to poor bioavailability and rapid metabolism. The core-shell nanoparticles designed in this way are effective carriers for resveratrol. As nanoparticles can better penetrate the tumour cells, the prepared compositions can be more effective in preventing and curing cancer.

References:

34. Lígia N.M. Ribeiro, Ana C.S. Alcântara, Margarita Darder, Pilar Aranda, Fernando M. Araújo-Moreira, Eduardo Ruiz-Hitzky. 2014. “Pectin-coated chitosan-LDH bionanocomposite beads for colon-targeted drug delivery”, *Int. J. Pharm.* 463, 1– 9.

<https://doi.org/10.1016/j.ijpharm.2013.12.035>

35. Ediana Paula Rebitski, Margarita Darder, Raffaele Carraro, Pilar Aranda and Eduardo Ruiz-Hitzky. 2020. Chitosan and pectin core–shell beads encapsulating metformin–clay intercalation compounds for controlled delivery. *New J. Chem.* 44, 10102—10110. DOI: 10.1039/c9nj06433h

2) Once placed on the topic, I see the core-shell structures here developed seem to present a much smaller size than the core-shell beads on the reports above signaled but here, my question is, why would it be necessary to create very small nanoparticles for controlled release of oral administered drugs? Actually, authors indicate in the text the possibility that the drug reach some organs in the body (they do not say which ones) but they do not offer any insights about how they expect the nanoparticles reach such organs. Or their system is intended for other type of administration?

Our response:

A nanosized drug can better penetrate the tumour cells and are more effective because of their large surface area. Orally administered drugs are mostly absorbed from the intestine, from where they can move to liver and then to the specific organ through systemic circulation.

- Secondly, I found other significant points for revision in the study that concerns to the following points:

1) What is the type of chitosan (molecular weight, as they are various) was used? How were prepared the solutions of different pH? How was the protocol to determine the efficiency of drug loading and what it was not introduced in the experimental section (there is some reference to it later in the results and discussion section (??))?

Our response:

Chitosan used was of molecular weight 3800-20000 Dalton. This information has been inserted in the Materials and Methods section of the revised manuscript.

The solutions of different pH were prepared using buffer capsules of three different pH values (4, 7 and 9). This information has been inserted in experimental section of the revised manuscript under the heading "Study of drug (Resveratrol) delivery from RCP1, RCP2 and RCP3".

As per your suggestion, we have introduced the protocol for determination of the efficiency of drug loading into the experimental section. As it is a common and widely used protocol,

detailed description was not included in the earlier manuscript. The following text has been inserted in the revised manuscript-

The amount of resveratrol loaded in the prepared core shell nanoparticles was determined using UV-visible spectrophotometer. For this, the sample was dissolved in ethanol and the concentration of resveratrol was determined using the calibration graph.

2) When described the protocol for the preparation of the core-shell nanoparticles authors use terms “solution” and “suspension” but it is hard to follow because they are not equivalent and sometimes are used to refer to the same system. Moreover, it must be noted that it is difficult to solve chitosan and in the text, there is no explanation of the protocol followed to do it. Usually, chitosan is solved in acid to stabilize it protonated but in alcohol is hard to solve it, how do authors proceed in the present case? In Table 1 is signaled a composition of chitosan/pectin in the various systems, how were they determined? I cannot see any description of protocol indicating how the composition was analyzed.

Our response:

Corrections have been made by replacing the word “solution” with “suspension” which is more appropriate in the revised manuscript.

Resveratrol-loaded chitosan-pectin core-shell nanoparticles were prepared using antisolvent precipitation and electrostatic deposition method with slight modification as reported previously (20). 5.0 mL of ethanolic ~~solution~~-suspension of resveratrol loaded chitosan was rapidly injected into 20 mL water (adjusted to pH 4.0) with the help of a syringe. The mixture was then continuously stirred at 900 rpm using a magnetic stirrer. This process led to the formation of a suspension of resveratrol loaded chitosan nanoparticles. The resulting dispersion was stirred for another 3 minutes, and ethanol was evaporated from the nanoparticle suspension by heating the solution in a water bath at 80°C for 15 minutes. Appropriate amount of water (adjusted to pH 4.0) was added to the ~~solution~~ suspension to compensate any decrease in volume due to the evaporation of ethanol. The dispersion was poured into 35 ml pectin solution in water with continuous stirring for 30 minutes. The resulting ~~solution~~ suspension was then centrifuged for half an hour and the settled residue was dried in vacuum desiccator for 24 hours in order to obtain the desired Resveratrol loaded Chitosan-pectin core-shell nanoparticles (RCP1)

As chitosan is difficult to be dissolved, few drops of acetic acid was added while preparing its solution. This text has been inserted in experimental section under heading “Loading of Resveratrol in Chitosan”.

In Table 1 compositions of chitosan/pectin actually indicate the ratio of the two biopolymers taken during the preparation of the core-shell nanoparticles and not after the nanoparticles were formed.

3) There are few details of the protocol related to the study of drug delivery. Moreover, there some small indication regarding the RCP1 system but none to the RCP2 and RCP3. The same applies to the protocol regarding the antioxidant activity, only conditions related to the RCP1 system have been signaled. In this part again authors describe they prepare a “methanolic solution of RCP1”, how is it possible to speak of solve the nanoparticles?

Our response

The drug delivery was studied in details with all the three preparations i.e., RCP1, RCP2 and RCP3. The details about RCP2 and RCP3 have been mentioned under the heading “Effect of shell thickness on the release efficiency of the drug” in the Results and Discussion section of the manuscript. Also please, refer to the figures 6b and 6c which show the drug release pattern for RCP2 and RCP3.

However, the antioxidant activity was checked for RCP1 only as this was the best drug carrier as evident from the drug release pattern. Our main focus was to compare the scavenging activity of encapsulated resveratrol with free resveratrol. This is why we conducted the study taking the best sample out of the three.

The part “methanolic solution of RCP1” has been corrected to methanolic suspension of RCP1.

4) The FTIR characterization is incorrectly used the term peak when actually the correct one is band (something similar was later use when addressing UV-vis spectra). Authors must take care and use appropriate nomenclature when described signal in spectroscopic studies.

Additionally, it is not clear the assignation of bands to the resveratrol in the RCP1 system as, apparently, the major components in the core-shell nanoparticles are the biopolymers. So, the more intense bands in the 1500-500 cm^{-1} region probably correspond to bands of the polymers instead that can show shifts in their frequencies due to the existence of different interactions than in the pure biopolymers. Authors should check carefully this part to be sure of the interpretation they afford

Our response:

Correction has been done in FT-IR characterization part by replacing the word “peak” with “band” as follows-

The FT-IR spectra of chitosan (figure 1a) showed a broad band at 3451 cm^{-1} for N-H symmetrical stretch overlapped with O-H stretching band. The **peak band** at 2925 cm^{-1} was due to C-H stretching. The Chitosan polymer shows characteristic band of amide I (C=O stretch) at 1637 cm^{-1} and amide II (C-N stretching and N-H bending) at 1542 cm^{-1} . The **peak band** for amide III (N-H deformation vibration coupled with C-N stretching) is observed at 1374 cm^{-1} (21). Another **peak band** at 1160 cm^{-1} is due to C-O-C stretching of the saccharide units. The **peak band** at 1050 cm^{-1} is for NH bending vibration.

The FT-IR Spectra of Pectin (figure 1 b) shows a broad **peak band** at 3410 cm^{-1} due to O-H stretching. The **peak band** at 2925 cm^{-1} is due to C-H stretching. The polymer shows a typical **peak band** at 1754 cm^{-1} due to the presence of ester carbonyl C=O group (22). **Peak Bands** at 1644 cm^{-1} and 1408 cm^{-1} correspond to the COO- group stretching (range $1600\text{-}1650\text{ cm}^{-1}$ for antisymmetric stretch and $1400\text{-}1450\text{ cm}^{-1}$ for symmetric stretch). Another **peak band** at 1102 cm^{-1} is observed due to the presence of ether linkage and glycosidic bond in pectin. An absorption **peak band** at 1060 cm^{-1} is observed due to C-H bending.

The band assignment has been checked and matched with earlier reported works.

4. Images in Fig.2 are of very poor quality and it is not possible to see the scale so I am not able to distinguish anything. Anyway, the indication the particles are of 23-32 nm seems to be hard to ascertain based on them. Additionally, the EDX seems to be point less when the three components in the system are organic compounds. What was it used for?

Our response:

Quality of Figure 2 has been improved.

The particle size (23-32 nm) was mentioned in the original images recorded and printed by the FESEM instrument. This was used as it is.

EDX image was recorded as a routine exercise while recording FESEM images and not with any other particular intention. However, the absence of other elements indicates the purity of the samples.

5. The FESEM study seems to be applied to determine and see the size, shape and structure of the nanoparticles but from the afforded images it seems that only the images in Figure 3a, corresponding to the RCP1 system look like core-shell nanoparticles, the images of the other systems show the presence of particles with not well define shapes and structures. By the way, images in Fig. 3b and 3c seem to be duplicate. From those images it is not possible to

see how the values of shell thickness and average diameter of the particles in Table 1 were established. Moreover, how it is possible to determine values with a precision of 0.01 nm from those images. From data in the Figure 4 it seems the particles are much larger than those authors said are the ones deduced from the TEM, at this point it is hard to believe that the agglomerates that suggest figures in DLS graphics were constituted by hundreds of nanoparticles (differences from 20 nm to > 1000 nm).

Our response:

For RCP2 and RCP3 HRTEM images were recorded which were somewhat different from FETEM images of RCP1. However, the enlarged images show the core-shell nanoparticles clearly. Not defined shapes and structures may be due to poor dispersion of the sample in the solvent during sample preparation for recording images in HRTEM.

The same figure was repeated as figure 3c and 3b by mistake. Correction has been done.

The diameter and thickness of core shell nanoparticles were determined manually using the scale provided in each image (after enlarging the image the nanometre scale was converted into mm scale and the measurements were performed). Moreover, Image J software was also used to recheck the correctness of the sizes measured manually.

The size obtained in DLS is the hydrodynamic radius which means the radius of the imaginary spheres formed when water molecules surround the core shell nanoparticles. This size is naturally much larger.

6. Interpretation of zeta potential is not clear, why authors say the values in system with ratio 1:15 indicated good stability, in relation to what? From this analysis is not clear if authors understand what is the information this technique affords and so their interpretation is totally unclear.

Our response:

The relation between the zeta potential value and the stability of the nanoparticles has been properly explained in the revised form. In our study, RCP2 having Chitosan: pectin in the ratio 1:15 showed higher zeta potential values compared to the other two. Thus, it can be predicted that the nanoparticles in RCP2 have higher stability compared to other two core-shell nanoparticles. Following changes have been made in the manuscript-

Nanoparticles have a surface charge that attracts a thin layer of ions of opposite charge to the nanoparticle surface. This double layer of ions travels with the nanoparticle as it diffuses throughout the solution. The electric potential at the boundary of the double layer is known as the Zeta potential of the particles. The zeta potential, which depends on the surface charge, is an important factor for the stability of nanoparticles in suspension and is also the major factor in the initial adsorption of nanoparticles onto the cell membrane. The values of zeta potential can range from +100 mV to -100 mV. Nanoparticles with Zeta Potential values greater than +25 mV or less than -25 mV typically have high degrees of stability. Dispersions with a low zeta potential value will eventually aggregate due to Van Der Waal inter-particle attractions (28, 29).

References:

28. Hunter, R. J. Zeta Potential in Colloid Science. Principles and Applications (Academic Press, 1981). ISBN: 9781483214085.

29. Honary, S. & Zahir, F. Effect of zeta potential on the properties of nano-drug delivery systems—a review (part 1). 2013. Trop. J. Pharm. Res. 12, 255–264.

7. There is a section named as “Particle yield and Reveratrol loading efficiency” which describe some experiments that should be better included as an experimental section although it actually does not introduce a complete protocol but just certain information of how the loading of resveratrol was estimated. The information is not complete and also introduces doubts about the efficiency of the method as it is indicated the system was centrifuged at 1000 rpm for 10 minutes and it is hard to believe that in this situation was possible to separate nanoparticles as smaller as 20 nm as authors claim they have. Other points in this study is what there are only results for the RCP1 system and not to the other ones.

Our response:

The mentioned part (protocol) has been shifted to experimental section with more information. The result section contains only the results. The best results were obtained from RCP1. The results from the other two were less. The best one is reported.

8. I will not introduce any comment on the drug delivery system study as far as I have not clear idea of what is the actual goal of the prepared systems, oral administration? “targets different organ sites” as signaled in the text? If it is for oral administration it is not possible to understand the experiments of release at different pH was not done in a sequential order as the one of the changes in the gastro-intestinal tract or why the release is kept till 30 hours when the maximum is much shorter. If the intended application is to introduce the nanoparticles in the organs then the essays have to be explained and designed more clearly.

Our response:

The actual goal of the prepared systems was oral administration. The drug delivery behaviour of the samples was checked at different pH and the details are already mentioned in Results and Discussion section under heading “Effect of pH on release efficiency of the drug from core shell nanoparticles” in the manuscript.

The study was conducted for a prolonged period of 30 hours because this is the approximate time for which a preparation can stay inside the GI tract of human body.

9. If the main conclusion of the works is just a “pectin coating can serve as an effective strategy to stabilize chitosan loaded nanoparticles as oral delivery vehicles” as main outcome of the study, then it must be signaled that this is not something new. In fact, previous results by other authors, as I had signaled above already reported on it, and therefore, other points should be strengthen to justify the study.

Our response:

The main outcome of the study is not at all the one mentioned here. Though clearly stated in the conclusion section of the manuscript, we would like to highlight again the major outcome of the study for quick reference.

1. Our focus was to design an effective carrier for resveratrol which can protect it from rapid metabolism and at the same time release the drug slowly for a sufficiently long time.
2. The antioxidant activity of the loaded resveratrol was found to be higher then the free resveratrol. This is a significant outcome of the study because encapsulation can prove to be an effective strategy to modify and magnify the biological activity.
3. As nanoparticles can better penetrate the tumour cells, the prepared compositions can be more effective in preventing and curing cancer.

We further state that the present work is not a repetition of earlier reported works but indicates a possible extension and application of earlier similar works.

10. Finally, it must be signaled the manuscript introduces diverse errata and typos that should be revised. Please, do not use capital letter when refers to common words such chitosan, pectin, etc.

Our response:

We have replaced the capital letters and revised to remove the typographic errors as much as possible.

Reviewer: 5

Comments to the Author(s)

The manuscript is very interesting concerning the resveratrol loaded chitosan-pectin core-shell nanoparticles as novel drug delivery vehicle for sustained release and improved antioxidant activities. Most of the questions have been addressed. The reviewer suggest the acceptance of this manuscript. Advices to the authors, but not strongly recommended, especially about the DLS characterization:

As far as self-assembly of polymeric molecules in different solvents is commonly studied by DLS and the aggregation process can be investigated using this method, Although the particle sizes of TEM must be smaller than that of DLS, but the intensity sizes were much too different from the results of TEM. As usual, the more uniform of the distribution of nanoparticles, the closer of the number, intensity and volume sizes were. Therefore I suggest the authors provide the volume (%) sizes of nanoparticle, which are more persuasive than the intensity size.

Our response:

The volume (%) sizes of nanoparticle in DLS are nearly same as was obtained for intensity sizes.

Appendix D

Response to the referee's comments

Journal Name: - RSOS (Royal Society Open Science)

Manuscript ID: RSOS-210784 R2

TITLE: Resveratrol Loaded Chitosan-Pectin Core-Shell Nanoparticles as Novel Drug Delivery Vehicle for Sustained Release and Improved Antioxidant Activities

Dear Sir,

Thank you for suggesting minor revisions to the manuscript. The manuscript has been modified based on the suggestions and comments made. Detailed corrections and explanations in manuscript are listed below point by point.

Comments:

Reviewer:

I have read with attention the response to my queries and the revised manuscript and I regret to express that some of the most relevant points I signaled in my previous report were not at all understood in some cases, and in other just skipped. Now at least I can understand better the goal of the study in relation to the target of delivery so, it makes clearer that some of my previous questions were not addressed in the revised manuscript that, by the way, introduces just a few changes. As I had pointed out in my previous report, the development of chitosan-pectin core-shell nanoparticles to produce a drug delivery system for oral administration of resveratrol is new, but the use of chitosan-pectin core-shell structures to produce drug delivery systems for oral administration of drugs is not new. Amongst the diverse reports in the I had signaled two examples (one selected because it was the first publication on the topic and the other one as a specific example in which the system is improved to oral delivery a drug that present problems for being adsorbed, as in the present case), for introducing the topic and to place the novelty of the here reported approach in context. **Therefore, this should be stressed in the introduction not somewhere else within the results and discussion section because the approach here used for oral drug delivery is not at all new even though it is now applied to delivery another specific drug, resveratrol, and the produced spheres are nanometric instead of micro or milimetric as in other approaches.**

Our response:

As suggested by the reviewer, the point has been shifted from Results and Discussion section to the Introduction section to make the novelty and the objective of the work clear to the readers.

Reviewer:

The second point that arisen some concerns to me is related to the precise mechanism of action of the produced drug delivery system. From the answer authors made to my first query, it looks that the produced nanospheres will be adsorbed at the intestine as they are and then will be “move to liver and then to the specific organ through systemic circulation” (sic). **This point is relevant and the manuscript doesn't offer proof it could work from a physiological point of view, is there any evidence with similar type of systems? In general, drug delivery systems for actions at specific location of tumors use to be delivery via intravenous to assure it can reach the target, some references to support they**

Our response:

Yes, the statement is based on the earlier reports of nanoparticle-based oral drug delivery systems and the cellular distribution pattern of the drugs. The following review and articles may be referred-

1. Susan Hua. Advances in Oral Drug Delivery for Regional Targeting in the Gastrointestinal Tract - Influence of Physiological, Pathophysiological and Pharmaceutical Factors. *Front. Pharmacol.*, 2020, volume 11, Article 524. <https://doi.org/10.3389/fphar.2020.00524>.
2. Miaorong Yu, Yiwei Yang, Chunliu Zhu, Shiyang Guo, Yong Gan, Advances in the transepithelial transport of nanoparticles, *Drug Discovery Today*, Volume 21, Issue 7, 2016, Pages 1155-1161, <https://doi.org/10.1016/j.drudis.2016.05.007>.
3. Jonas Reinholza, Katharina Landfester and Volker Mailander. The challenges of oral drug delivery via nanocarriers. *Drug Delivery*, 2018, 25:1, 1694-1705, DOI: 10.1080/10717544.2018.1501119.
4. Anne des Rieux, Eva G.E. Ragnarsson, Elisabet Gullberg, Veronique Pr ´ eat ´, Yves-Jacques Schneider, Per Artursson. *European Journal of Pharmaceutical Sciences*, 25 (2005) 455–465.

Reviewer:

Regarding my other queries, some of them have been explained and revised except other ones that I consider relevant and, mainly they have been just ignored:

- I insist, the EDX study is pointless when the three components in the system are organic compounds. What were the expected “other elements” to be concerned in relation to the “purity of the samples”?

Our response:

Though it was mentioned that EDX study was carried out as a routine procedure along with SEM and not with any particular objective, we are removing the EDX part from the manuscript to honour the reviewer's suggestion.

By “other elements” we actually meant heavy metal contaminants frequently found in chitosan samples prepared from Shrimp Shells.

Reviewer:

- I do not see any response in relation to the proposed protocol of separation of nanoparticles using just centrifugation at 1000 rpm for 10 minutes.

Our Response:

Please note that we have never claimed to have separated nanoparticles using just centrifugation at 1000 rpm for 10 minutes. Please go through the manuscript section **Particle yield and Resveratrol loading efficiency**. We have mentioned that centrifugation at 1000 rpm for 10 minutes was done only **in order to separate out any large particles** from the freshly prepared colloidal dispersions of the synthesized core shell nanoparticles RCP1. **We know that this condition cannot make the very small nanoparticles settle down.**

In “**Preparation of resveratrol loaded chitosan–pectin core-shell nanoparticles**” section it was clearly stated that the resulting suspension was centrifuged **for half an hour**. However, the centrifugation speed (3000 rpm) was not mentioned. For the convenience of the readers, the speed (3000 rpm) which was applied during separation of nanoparticles is included now in the revised manuscript.

Reviewer:

- The study of drug delivery at 3 pH but without a sequential protocol similar to the gastrointestinal tract is just informative but not conclusive when the intended application is a drug for oral administration. The results could inform of the stability and possible release in the different conditions but, to be more realistic, it is necessary to see what happens when the same system evolves from one to other pH medium.

Our Response:

The study was conducted at three different pH levels and not in a sequential order. The primary reason behind this is to start with the same initial concentration every time so that the influence of the pH can be made distinct (without being influenced by other variables like time, temperature etc.). We have done extensive literature survey before designing the release study and have found this approach adopted by several earlier workers. The findings of the experiment are informative and could be useful for the design and fabrication of polymeric nanoparticles.

We thank you for your suggestion of applying a sequentially changing pH. Such a study can be interesting and we shall try to incorporate that in our future publications.

Reviewer:

Additionally, I am quite surprised that, with the small size of the nanoparticles used in the study, after 30 hours the release was not complete, as the biopolymers may dissolve, at least partially. Is there any evidence of the same of the nanoparticles after the release study?

Our Response:

Yes, there are many published reports where the release of the drug was studied up to 24 hours, 30 hours and even longer. (A list is provided below). In fact, the biopolymer composite is prepared with the aim that the drug release becomes slow and sustainable. The biopolymers dissolve slowly and the drug release is continued till it completely disintegrates.

1.Zhou L, Huang Z, Yang S, et al. Preparation of ICA-loaded mPEG-ICA nanoparticles and their application in the treatment of LPS-induced H9c2 cell damage. *Nanoscale Res Lett.* 2021;16(1):155 (16 pages).

2.Puri S, Kallinteri P, Higgins S, Hutcheon GA, Garnett MC. Drug incorporation and release of water - soluble drugs from novel functionalised poly (glycerol adipate) nanoparticles, *Journal of Controlled Release*, 2008; 125(1), Pages 59-67.

3.Men WF, Zhu PY, Dong SY, Liu WK, Zhou K, Bai Y, Liu XL, Gong SL, Zhang, SG, Layer-by-layer pH-sensitive nanoparticles for drug delivery and controlled release with improved therapeutic efficacy in vivo, *Drug Delivery*, 2020; 27(1), Pages 180-190.

Reviewer:

Finally, I have still concerns in relation to the statement that “the prepare nanoparticles can better penetrate the tumor cells, etc.” authors introduced for answering my query related to the conclusions part. From the introduced results, the system could be promised for it but the manuscript does not show conclusive evidence of it. Therefore, I would be more careful at this respect.

Our Response:

The fact that the nanoparticles can penetrate tumour cells better is well established. Several studies have already reported that the tumour penetration power of nanoparticles is much better than larger sized particles. Tumour penetration study has not been carried out here and the authors do not have the necessary expertise to carry out such work. The statement is made on the basis of earlier reports as listed below-

- 1 Susan Hua. Advances in Oral Drug Delivery for Regional Targeting in the Gastrointestinal Tract - Influence of Physiological, Pathophysiological and Pharmaceutical Factors. *Front. Pharmacol.*, 2020, volume 11, Article 524. <https://doi.org/10.3389/fphar.2020.00524>.